# Improving Bilinear RNNs with Closed-loop Control

**Jiaxi Hu**[1]    **Yongqi Pan**[1] *    **Jusen Du**[2]    **Disen Lan**[2]    **Xiaqiang Tang**[1]    **Qingsong Wen**[3]
**Yuxuan Liang**[1]✉    **Weigao Sun**[2]✉

[1]The Hong Kong University of Science and Technology (Guangzhou)
[2]Shanghai AI Laboratory    [3]Squirrel Ai Learning, USA

## Abstract

Recent efficient sequence modeling methods such as Gated DeltaNet, TTT, and RWKV-7 have achieved performance improvements by supervising the recurrent memory management through Delta learning rule. Unlike previous state-space models (e.g., Mamba) and gated linear attentions (e.g., GLA), these models introduce interactions between the recurrent state and the key vector, structurally resembling bilinear systems. In this paper, we first introduce the concept of Bilinear RNNs with a comprehensive analysis on the advantages and limitations of these models. Then, based on closed-loop control theory, we propose a novel Bilinear RNN variant named Comba, which adopts a scalar-plus-low-rank state transition, with both state feedback and output feedback corrections. We also implement a hardware-efficient chunk-wise parallel kernel in Triton and train models with 340M/1.3B parameters on large-scale corpus. Comba demonstrates superior performance and computation efficiency in both language and vision modeling.

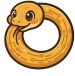 https://github.com/fla-org/flash-linear-attention.

> *"Learning without thinking misleads."*
>
> *– Confucius, 479 BC*

## 1   Introduction

Autoregressive Transformers [95] have become a foundation of modern AI, primarily due to the efficient parallel computation made possible by softmax-based self-attention. This mechanism enables effective memory scaling by directly appending `key` and `value` vectors into the KV cache, which contributes to strong performance on tasks such as in-context learning and long-context retrieval. However, this design also comes with challenges, including quadratic time complexity and unbounded memory growth during inference [59], which limits the model's scalability for long-sequence tasks. To address these, numerous improvements have been introduced, including sliding window attention [8, 25], sparse attention techniques [64, 111, 103], and efficient KV cache management [59].

Meanwhile, efficient sequence-mixing approaches such as gated linear attention [112, 79, 22, 107, 105, 5] and selective state space models (SSMs) [38, 26, 84] offer a compelling alternative. These models aim to establish a linear *key-value associative memory* [105, 34] register with constant states and data-(in)dependent gating ($\boldsymbol{\alpha}$, $\boldsymbol{\beta}$), as presented by $\boldsymbol{S}_t = \boldsymbol{\alpha}_{(t)}\boldsymbol{S}_{t-1} + \boldsymbol{\beta}_{(t)}\boldsymbol{v}_t\boldsymbol{k}_t^\mathsf{T}$. The inherent recurrent structure of these models enables them to maintain constant memory overhead and $\mathcal{O}(1)$ time complexity during inference. Despite these models originating from distinct theoretical frameworks, for example, early linear attentions [53, 82] like Linformer [98] attempt to reformulate quadratic attention computation as $\boldsymbol{O} = \phi(\boldsymbol{Q})(\phi(\boldsymbol{K})^\mathsf{T}\boldsymbol{V})$ with kernel mapping $\phi$, while original SSMs [39, 42] like S4 [41] intent to parameterize a continuous dynamical system to a discrete form, recent literature [22, 7, 52, 6] have unified these models to the concept of Linear RNNs.

---

✉Corresponding authors (yuxliang@outlook.com, sunweigao@outlook.com). * Work done during Yongqi Pan's internship at Hong Kong University of Science and Technology (Guangzhou).

39th Conference on Neural Information Processing Systems (NeurIPS 2025).

Table 1: **Memorizing Mechanisms in state-of-the-art Sequence Modeling Methods.** Softmax Attention ensures precise memory storage, while sliding window attention constrains storage space. Linear RNNs (lines 3-6) use data-(in)dependent gate for unsupervised memory management, with Mamba2 approximating the forget gate $\alpha$ to 1, forcing the model to forget. Bilinear RNNs (lines 7-9) is no longer a linear *key-value memory register* $\boldsymbol{S}_{t+1} = (\boldsymbol{\alpha}, \boldsymbol{\beta})@(\boldsymbol{S}_t, \boldsymbol{k}_t^\mathsf{T} \boldsymbol{v}_t)^\mathsf{T}$, which supervise the management process based on the Delta learning rule – effectively equivalent to minimizing a Stochastic Gradient Descent $\nabla_S \|\boldsymbol{S}_t \boldsymbol{k}_t - \boldsymbol{v}_t\|^2$. When ignoring layer normalization and residual components, TTT-Linear can also be categorized as a bilinear RNN. As modern Nonlinear RNNs, MIRAS and its variants (e.g., TTT-MLP, Titans [7], Lattice [52]) have stronger expressiveness because of the nonlinearity $g$, high-order optimizations, and MLP-based deep memory, but are limited by the chunk-wise parallelism (§3). Our proposed Comba further improves Bilinear RNNs by closed-loop control.

| Model | Memorizing with Gate | Optimization Objective $\mathcal{L}$ |
|---|---|---|
| *Softmax Attention: Conductivist-based infinite key-value associative memory registers* | | |
| SA [95] | $\boldsymbol{S}_t = \boldsymbol{S}_{t-1}.\,\mathrm{append}(\boldsymbol{k}_t, \boldsymbol{v}_t)$ | $\sum_{i=1}^t \exp(\boldsymbol{q}_t^\mathsf{T}\boldsymbol{k}_i)\|\boldsymbol{v} - \boldsymbol{v}_i\|^2$ [97] |
| SWA [8] | $\boldsymbol{S}_t = \boldsymbol{S}_{t-1}.\,\mathrm{append}(\boldsymbol{k}_t, \boldsymbol{v}_t).\,\mathrm{drop}(\boldsymbol{k}_{t-M}, \boldsymbol{v}_{t-M})$ | $\sum_{i=t-M}^t \exp(\boldsymbol{q}_t^\mathsf{T}\boldsymbol{k}_i)\|\boldsymbol{v} - \boldsymbol{v}_i\|^2$ |
| *Linear RNNs: Inductivist-based finite key-value associative memory registers* | | |
| LA [98] | $\boldsymbol{S}_t = \boldsymbol{S}_{t-1} + \boldsymbol{v}_t \boldsymbol{k}_t^\mathsf{T}$ | $-\langle \boldsymbol{S}_t \boldsymbol{k}_t, \boldsymbol{v}_t \rangle$ |
| GLA [107] | $\boldsymbol{S}_t = \boldsymbol{S}_{t-1}\mathrm{diag}(\boldsymbol{\alpha}_t) + \boldsymbol{v}_t \boldsymbol{k}_t^\mathsf{T}$ | $-\langle \boldsymbol{S}_t \boldsymbol{k}_t, \boldsymbol{v}_t \rangle + \frac{1}{2}\left\|\sqrt{\mathrm{diag}(\boldsymbol{1} - \boldsymbol{\alpha}_t)}\boldsymbol{S}_t\right\|_F^2$ |
| HGRN2 [79] | $\boldsymbol{S}_t = \boldsymbol{S}_{t-1}\mathrm{diag}(\boldsymbol{\alpha}_t) + \boldsymbol{v}_t(\boldsymbol{1} - \boldsymbol{\alpha}_t)^\mathsf{T}$ | $-\langle \boldsymbol{S}_t(\boldsymbol{1} - \boldsymbol{\alpha}_t), \boldsymbol{v}_t \rangle + \frac{1}{2}\left\|\sqrt{\mathrm{diag}(\boldsymbol{1} - \boldsymbol{\alpha}_t)}\boldsymbol{S}_t\right\|_2^2$ |
| Mamba2 [26] | $\boldsymbol{S}_t = \alpha_t^{\sim 1}\boldsymbol{S}_{t-1} + \beta_t^{\sim 0}\boldsymbol{v}_t \boldsymbol{k}_t^\mathsf{T}$ | $-\beta_t \langle \boldsymbol{S}_t \boldsymbol{k}_t, \boldsymbol{v}_t \rangle + \frac{1}{2}\left\|\sqrt{\boldsymbol{1} - \alpha_t}\boldsymbol{S}_t\right\|_2^2$ |
| *Bilinear RNNs: Moving beyond linear key-value memory registers with memory correction* | | |
| G-DeltaNet [105] | $\boldsymbol{S}_t = \boldsymbol{S}_{t-1}\alpha_t^{\sim 1}\left(\boldsymbol{I}_t - \beta_t \boldsymbol{k}_t \boldsymbol{k}_t^\mathsf{T}\right) + \beta_t \boldsymbol{v}_t \boldsymbol{k}_t^\mathsf{T}$ | $\frac{1}{2}\alpha_t\beta_t\left\|\frac{1}{\alpha_t}\boldsymbol{v}_t - \boldsymbol{S}_t \boldsymbol{k}_t\right\|^2 + \frac{1}{2}\left\|\sqrt{\boldsymbol{1} - \alpha_t}\boldsymbol{S}_t\right\|_2^2$ |
| RWKV7 [73] | $\boldsymbol{S}_t = \boldsymbol{S}_{t-1}(\mathrm{diag}(\boldsymbol{\alpha}_t) - \beta_t \hat{\boldsymbol{k}}_t \hat{\boldsymbol{k}}_t^\mathsf{T}) + \boldsymbol{v}_t \tilde{\boldsymbol{k}}_t^\mathsf{T}$ | $\frac{1}{2}\beta_t\left\|\frac{1}{\beta_t}\boldsymbol{v}_t - \boldsymbol{S}_t \boldsymbol{k}_t\right\|^2 + \frac{1}{2}\left\|\sqrt{\mathrm{diag}(\boldsymbol{1} - \boldsymbol{\alpha}_t)}\boldsymbol{S}_t\right\|_2^2$ |
| **Comba (ours)** | $\boldsymbol{S}_t = \boldsymbol{S}_{t-1}\left(\alpha_t^{\sim 1} - \beta_t^\downarrow \boldsymbol{k}_t \boldsymbol{k}_t^\mathsf{T}\right) + \beta_t^\uparrow \boldsymbol{v}_t \boldsymbol{k}_t^\mathsf{T}$ | $\frac{1}{2}\beta_t\|\boldsymbol{v}_t - \boldsymbol{S}_t \boldsymbol{k}_t\|^2 + \frac{1}{2}\left\|\sqrt{\boldsymbol{1} - \alpha_t}\boldsymbol{S}_t\right\|_2^2 - \langle \boldsymbol{q}_t, d\boldsymbol{k}_t \rangle$ |
| *(Modern) Nonlinear RNNs: Stronger expressiveness and memory capacity, but limited in chunk-wise parallelism.* | | |
| TTT-MLP [90] | $\boldsymbol{S}_t(\cdot) = \boldsymbol{S}_{t-B}(\cdot) - \sum_{i=1}^B \beta_i \nabla_S \|\psi(\boldsymbol{S}_{t-B}(\boldsymbol{k}_i)) - \boldsymbol{v}_i\|^2$ | $\beta_i \|\boldsymbol{v}_i - \psi(\boldsymbol{S}_j(\boldsymbol{k}_i))\|^2$ |
| MIRAS [6] | $\boldsymbol{S}_t = \alpha_t \boldsymbol{S}_{t-1} - \beta_t \nabla_S \|g(\psi(\boldsymbol{S}_{t-1}), \boldsymbol{k}_t) - \boldsymbol{v}_t\|_p^p$ | $\beta_t \|\boldsymbol{v}_t - g(\psi(\boldsymbol{S}_t), \boldsymbol{k}_t)\|_p^p + \frac{1}{2}\left\|\sqrt{\mathrm{diag}(\boldsymbol{1} - \boldsymbol{\alpha}_t)}\boldsymbol{S}_t\right\|_2^2$ |

$\boldsymbol{S}_t$ denotes memory, while $\boldsymbol{k}_t, \boldsymbol{v}_t$ are key-value pairs. $\alpha_t^{\sim 1}$ denotes close to 1. $\beta_t^\downarrow, \beta_t^\uparrow$ represent smaller/bigger factors. Early Nonlinear RNNs, such as LSTM [46] and GRU [21] are omitted on the table. Softmax attention can also be interpreted as an L1 loss with a kernel function [114].

The data-dependent gating in Linear RNNs provides a dynamic memory management similar to the adaptive, weighted information fusion in softmax attention. This allows such models to selectively update and retain relevant information, enabling Linear RNNs to serve as practical replacements for Transformers in many downstream tasks [48, 62, 49]. However, this mechanism remains heuristic; that is, the model lacks a criterion for determining which memories to forget, and all key-value associations are forgotten uniformly, rendering the process less targeted and efficient [105].

To address this, recent models such as DeltaNet [108, 105], RWKV-7 [73], and TTT [90] have advanced state transition to generalized Householder transformations [51], enhancing model's learning capacity and enabling supervised memory control via the Delta learning rule [101]. These architectural shifts foster richer interactions between the internal state $\boldsymbol{S}$ and the input information $\boldsymbol{k}$, moving beyond simple linear *key-value memory registers* and resembling bilinear dynamics. Consequently, we refer to these models as Bilinear RNNs in this paper. Further improvements [7, 52, 6] have built upon TTT by introducing higher-order nonlinear optimization or MLP-based deep memory, giving rise to modern Nonlinear RNNs, which enhance expressiveness but sacrifice the ability to perform chunk-wise parallelism over the sequence. In summary, research in this field is still in its early stages, and there remain open challenges in achieving a good balance between model expressiveness and hardware-efficient implementation during pretraining.

In this paper, we make the following contributions:

- We summarize the progress of efficient sequence modeling methods in Table 1, and highlight the core design principles behind recent advances within the concept of Bilinear RNNs. (Section 2)
- Inspired by closed-loop control theory, we propose a novel Bilinear RNN architecture named ***Comba***. Unlike previous models, Comba features a scalar-plus-low-rank (SPLR) state transition and applies feedback control to the query vector during the output. We further develop a hardware-

friendly implementation of Comba using chunk-wise parallelism in Triton [93], which achieves a 40% speed improvement in forward propagation compared to Gated-DeltaNet. (Section 3)

- We pretrain models with 340M and 1.3B parameters and evaluate their performance across a range of tasks, including language modeling and vision tasks. Extensive ablation studies are conducted to assess the impact of key components of Comba. (Section 4)

## 2 Preliminary & Related Works

### 2.1 Linear RNNs

Unlike autoregressive Transformers that store all contextual information in KV cache, linear RNNs compress highly abstract knowledge into a fixed-size state for generalization, structurally resembling energy-based models [56], i.e., Hopfield networks [65, 31] and neural Hebbian learning systems [18]. Early models like Linformer [98], S4 [41] and RetNet [91] lack effective, data-dependent memory control, resulting in inferior performance to softmax attention. Later models like Mamba [38] and GLA [107] address this by introducing a dynamic, projection-based gating, yielding substantial improvements. Formally, these models are linear register systems with *key-value associative memory*, where the memory is written by the forgetting/input gates $(\boldsymbol{\alpha}, \boldsymbol{\beta})$ and retrieved via query-based read:

$$\boldsymbol{S}_t = (\boldsymbol{\alpha}_t, \boldsymbol{\beta}_t)@(\boldsymbol{S}_{t-1}, \boldsymbol{k}_t^\mathsf{T}\boldsymbol{v}_t)^\mathsf{T} \quad \textit{(Write)}, \qquad \boldsymbol{o}_t = \boldsymbol{S}_t\boldsymbol{q}_t \quad \textit{(Read)}. \tag{1}$$

Due to differences in theoretical foundations and development trajectories, linear RNNs have evolved into two main implementation paradigms: (i) linear attentions (LAs) [112, 79, 22, 107, 105, 5] and (ii) state space models (SSMs) [72, 38, 39], we summarize their key distinctions in Table 2:

i) **State size**: SSMs like Mamba2 [26] adopt a fixed state expand dimension of 128, resulting in a state size of 256D. Whereas in LAs, the state size is determined by the dimension of key/value heads (typically 64). In practice, Mamba2 offers a higher capacity [20], which underpins its advantages in retrieval tasks and hybrid architectures [88, 89].

Table 2: Comparison between SSM and LA families with the head number H.

| Component | SSMs (Mamba2) | LAs |
|---|---|---|
| Input value | $\boldsymbol{u}_t \in \mathbb{R}^{2D/H}$ | $\boldsymbol{v}_t \in \mathbb{R}^{dv/H}$ |
| State expand | $\bar{\boldsymbol{B}}_t \in \mathbb{R}^{128}$ | $\boldsymbol{k}_t \in \mathbb{R}^{dk/H}$ |
| Output | $\bar{\boldsymbol{C}}_t \in \mathbb{R}^{128}$ | $\boldsymbol{q}_t \in \mathbb{R}^{dk/H}$ |
| State size | $\boldsymbol{x}_t : 256 \times D$ | $\boldsymbol{S}_t : \frac{dk \times dv}{H}$ |
| MLP | ✗ | ✔ |
| Mode | Multi-value | Multi-head |

ii) **Parameter composition**: SSMs like Mamba2 resembles a multi-value attention mechanism [26], where the input projection $\bar{\boldsymbol{B}}$ ($\boldsymbol{k}$) and output projection $\bar{\boldsymbol{C}}$ ($\boldsymbol{q}$) are shared across all value heads $\boldsymbol{u}$ ($\boldsymbol{v}$). Additionally, state space models omit the feedforward network and instead double both the input dimension and the number of layers to increase model capacity. In this paper, we empirically follow the linear attention design but keep the head dimension at 256 to match the state size of Mamba2. A detailed architectural ablation is in §4.2.

**Chunk-wise Parallel** Although Linear RNNs achieve a favorable pretraining time complexity of $\mathcal{O}(LD^2)$, they often slower than softmax attention with $\mathcal{O}(L^2D)$ complexity on shorter sequences. This is mainly due to the fact that current hardware is highly optimized for `matmul` operations, which limits the efficiency of linear recurrence, necessitating additional training optimizations. S4 [41] introduces a complex diagonal-plus-low-rank design using the Cauchy kernel, which is later simplified in DSS [43] and S4D [40] through complex and real diagonal approximations to employ Vandermonde-based convolutions. Other models like S5 [84] and Mamba [38] leverage the Blelloch scan algorithm [15] to cache intermediate results and speed up recurrent computation. Recent methods inspired by FlashAttention [25], including Lightning-Attns [77, 78, 58], GLA [107], and Mamba2 [26], introduce inter-chunk recurrence combined with intra-chunk parallelism to fully utilize matrix compute throughput. A basic formulation using chunk size $C$ can be expressed as:

$$\boldsymbol{S}_{[t+1]} = \boldsymbol{S}_{[t]} + \boldsymbol{V}_{[t]}^\mathsf{T}\boldsymbol{K}_{[t]} \in \mathbb{R}^{D \times D}, \quad \boldsymbol{O}_{[t]} = \boldsymbol{Q}_{[t]}\boldsymbol{S}_{[t]}^\mathsf{T} + (\boldsymbol{Q}_{[t]}\boldsymbol{K}_{[t]}^\mathsf{T} \odot \mathrm{Mask}_{[t]})\boldsymbol{V}_{[t]} \in \mathbb{R}^{C \times D}. \tag{2}$$

### 2.2 Bilinear RNNs and Beyond

In a neural memory perspective [34], effective memory management remains a central challenge. Unlike Hebbian learning rule [18], which relies on reinforcement-based memory updates, Delta learning rule [75, 101] focuses on supervised memory control and has been extensively explored in various works, such as fast weight programs [50, 81] and Meta-learning [68, 67]. (Gated-)DeltaNet

[108, 105] employs it as a memory correction $\boldsymbol{v}_t^{\text{new}} = \boldsymbol{v}_t - \boldsymbol{S}_{t-1}\boldsymbol{k}_t$, and based on Eq. 2, introduces an efficient chunk-wise parallel algorithm for hardware-efficient training in sequence modeling:

$$\boldsymbol{S}_t = \boldsymbol{S}_{t-1} - \beta_t(\boldsymbol{S}_{t-1}\boldsymbol{k}_t - \boldsymbol{v}_t)\boldsymbol{k}_t^{\mathsf{T}} = \boldsymbol{S}_{t-1}(\boldsymbol{I} - \beta_t\boldsymbol{k}_t\boldsymbol{k}_t^{\mathsf{T}}) + \beta_t\boldsymbol{v}_t\boldsymbol{k}_t^{\mathsf{T}}, \qquad \boldsymbol{o}_t = \boldsymbol{S}_t\boldsymbol{q}_t. \qquad (3)$$

These models are no longer a linear *key-value memory register* as in Eq. 1; instead, the interaction between the state $\boldsymbol{S}$ and the input $\boldsymbol{k}$ introduces a bilinear term $\boldsymbol{Sk}$, this resembles a affine bilinear system[1] [17, 113, 99, 70]. Accordingly, models that are similar to the updating rule in Eq. 3 can be classified as ***Bilinear RNNs***, and their key strengths are summarized as follows:

**Supervised Memory Management**   As shown in Fig. 1, the Householder transform $(\boldsymbol{I} - \beta_t\boldsymbol{k}_t\boldsymbol{k}_t^{\mathsf{T}})$ generated by the Delta rule defines a mirror transform, effectively reflecting stored memories across a hyper-plane orthogonal to $\boldsymbol{k}_t$. Given a memory $\boldsymbol{S} \in \mathbb{R}^{D \times D}$ with at most $D$ orthogonal memory patterns, when the sequence length $t > D$, to avoid memory conflicts, this reflection attenuates components of stored memories $\{\boldsymbol{S}_{t-1}^i\}_{i=1}^D$ in non-orthogonal directions based on factor $\beta_t$. This mechanism implicitly enforces orthogonal memory management, enabling the

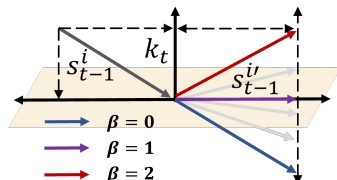

Figure 1: Householder transform as mirror transform with factor $\beta$.

model to preserve $D$ distinguishable historical memories over time. In this paper, Comba modifies the state transition to $(\alpha_t - \beta_t\boldsymbol{k}\boldsymbol{k}^{\mathsf{T}})$, which is a scalar-plus-low-rank form (SPLR), enabling more flexible supervision. To some extent, this process can be seen as a Schmidt orthogonalization [57] or a rotation operation [86] on the KV cache.

**Richer Expressive Power**   Linear RNNs generally approximate the dense state transition matrix [66, 46] with a diagonal matrix or scalar [40, 26], significantly reducing computational overhead but at the cost of expressiveness [1]. While the additional low-rank terms of the state transition in Eq. 3 improve the model's expressiveness while preserving tractability for efficient parallelization.

Building on this, Gated-DeltaNet [105] introduces a global scalar forgetting gate on the state, and Delta-Product [83] explore a multi-step Householder transform $\prod_{j=1}^n(\boldsymbol{I} - \beta_{t,j}\boldsymbol{k}_{t,j}\boldsymbol{k}_{t,j}^{\top})$, enabling a smooth interpolation between purely diagonal and fully dense transitions. RWKV7 [73] improves the IPLR form to diagonal-plus-low-rank (DPLR) form, which aligns with insights from HiPPO theory [39, 42], which shows that all orthogonal polynomial projection matrices can be decomposed into DPLR components. Our proposed Comba adopts a scalar-plus-low-rank (SPLR) form. Empirically, we find that the scalar is sufficiently expressive (similar to the empirical simplification from Mamba1 to Mamba2) and offers significant pretraining acceleration over RWKV-7.

From another perspective, models such as TTT [90] separate the model weights into inner and outer components, updating the inner memory directly via stochastic gradient descent (SGD) [16].

$$\boldsymbol{S}_t = \boldsymbol{S}_{t-B} - \sum_{i=1}^B \beta_i \nabla_{\boldsymbol{S}} \|\text{LayerNorm}(\boldsymbol{S}_{t-B}\boldsymbol{k}_i) - \boldsymbol{v}_i\|^2, \qquad \boldsymbol{o}_t = \boldsymbol{S}_t\boldsymbol{q}_t, \qquad (4)$$

where the state $\boldsymbol{S}$ can be parameterized by a matrix or a two-layer MLP to increase memory capacity. When mini-batch $B = 1$ and ignoring normalization operation, this reduces to the update rule in Eq. 3, and when adopting MLP-based deep memory, the model transitions into a modern form of Nonlinear RNN. Subsequently, Titans [7] introduces a data-dependent state decay and momentum, while models like MIRAS [6] and Lattice [52] enhance $\boldsymbol{S}$ using nonlinear higher-order optimization, e.g., sign function or high-order derivatives. However, there is no free lunch: these models rely on mini-batch gradient descent approach to compensate for the sequence-level parallelism barrier introduced by their structural complexity, but empirical results show that a mini-batch of 1 (similar to Eq. 3, which is a first-order approach) remains optimal [90], especially in language modeling. Comba retains the bilinear form and offers an efficient chunk-wise parallel optimization.

## 3   Bilinear RNNs in Closed-loop Control

Unlike perspectives from neural memory or optimization, this work revisits Bilinear RNNs through the lens of control theory [23, 24, 87]. As shown in Table 3, linear RNNs such as Mamba2 and

---

[1]Bilinear systems are linear with respect to state and input individually, but nonlinear overall due to the product term (e.g., $\boldsymbol{Sk}$). They are regarded as a special class of nonlinear systems that preserve controllability.

Table 3: Update rules in a control/neural memory perspective, with feedback $\boldsymbol{P}(\cdot)$ and scalar factor $d$.

| Option | Open-loop Control (Mamba2) | Close-loop Control (Comba) | Gated Delta Rule |
|---|---|---|---|
| *Input / Memorize* | $\boldsymbol{S}_t = \alpha_t \boldsymbol{S}_{t-1} + \beta_t \boldsymbol{v}_t^{\text{new}} \boldsymbol{k}_t^\mathsf{T}$ | $\boldsymbol{S}_t = \alpha_t \boldsymbol{S}_{t-1} + \beta_t \boldsymbol{v}_t^{\text{new}} \boldsymbol{k}_t^\mathsf{T}$ | $\boldsymbol{S}_t = \alpha_t \boldsymbol{S}_{t-1} + \beta_t \boldsymbol{v}_t^{\text{new}} \boldsymbol{k}_t^\mathsf{T}$ |
| *Feedback / Reflect* | N/A | $\boldsymbol{v}_t^{\text{new}} = \boldsymbol{v}_t - \boldsymbol{P}_t(\boldsymbol{S}_{t-1})$ | $\boldsymbol{v}_t^{\text{new}} = \boldsymbol{v}_t - \alpha_t \boldsymbol{S}_{t-1} \boldsymbol{k}_t$ |
| *Output / Recollect* | $\boldsymbol{o}_t = \boldsymbol{S}_t \boldsymbol{q}_t$ | $\boldsymbol{o}_t = \boldsymbol{S}_t \boldsymbol{q}_t - d\boldsymbol{P}_t(\boldsymbol{S}_t)$ | $\boldsymbol{o}_t = \boldsymbol{S}_t \boldsymbol{q}_t$ |

GLA are generally viewed, to some extent, as open-loop control systems, where the output does not provide feedback to influence the control behavior. In contrast, another class of systems, known as closed-loop control systems [45], incorporates negative feedback to enhance the adaptability of the system and allows it to handle more complex dynamic tasks. According to Wikipedia, a closed-loop controller should use feedback to control states or outputs of a dynamical system. So in this paper, Comba adopts a two-stage feedback strategy: the input information $\boldsymbol{v}_t$ is first corrected via state-based feedback $\boldsymbol{P}_t(\cdot)$, and the output is similarly refined using the same feedback mechanism. Compared to directly output correction methods, e.g., use $\boldsymbol{q}_t$ to compute $\boldsymbol{v}_t^{\text{new}}$, this approach is generally considered more robust and better suited for parallel computation (as it only modifies $\boldsymbol{q}_t$ at the output stage without involving it in state updates). From this perspective, models such as TTT, DeltaNet, and RWKV-7 incorporate only the first-stage state feedback correction.

**Feedback Parameterization**    Similar to the optimizer perspective, the feedback $\boldsymbol{P}_t(\cdot)$ in closed-loop control can be either linear and first-order, or nonlinear and higher-order. However, two major challenges arise: (i) adopting nonlinear or high-order optimization techniques, as in TTT, Lattice, or MIRAS, will hinder chunk-wise parallelism; and (ii) recurrent models suffer from the well-known issue of exponential gradient explosion [74] during training. If we follow DeltaNet using first-order feedback but initialize a new vector [106] to interact with the state, it becomes difficult to ensure that the spectral radius of the state transition matrix remains below one. Therefore, considering these factors, Comba follows previous models using the $\boldsymbol{k}$ vector to interact with the state $\boldsymbol{S}$, while introducing a special treatment of feedback strength detailed in the following sections. For a standard Comba with forget gate $\alpha_t$, state feedback factor $\tilde{\beta}_t$, input gate $\beta_t$, and output feedback factor $d$:

$$\boldsymbol{S}_t = \underbrace{\boldsymbol{S}_{t-1}(\alpha_t - \tilde{\beta}_t \boldsymbol{k}_t \boldsymbol{k}_t^\mathsf{T})}_{\text{State correction}} + \beta_t \boldsymbol{v}_t \boldsymbol{k}_t^\mathsf{T} \quad \in \mathbb{R}^{dv \times dk}, \qquad \boldsymbol{o}_t = \underbrace{\boldsymbol{S}_t(\boldsymbol{q}_t - d\boldsymbol{k}_t)}_{\text{Output correction}} \quad \in \mathbb{R}^{dv} \quad (5)$$

Compared to previous Bilinear RNNs, Comba exhibits the following structural differences:

**Scalar Plus Low-Rank (SPLR)**    In practice, the scalar gating will be limited to the interval $(0,1)$, and $\|\boldsymbol{k}_t\| = 1$ (L2 Norm). DeltaNet formulates the state transition in an IPLR structure, while RWKV-7 improves expressiveness via a DPLR extension with LoRA-style [47] diagonal initialization, i.e., $\text{diag}(\boldsymbol{w}_t) - \beta_t \boldsymbol{k}_t \boldsymbol{k}_t^\mathsf{T}$. However, our results indicate that the low-rank form will impair model capacity.

Table 4: State Transition for Comba Variants.

| Version | State transition | Eigenvalues |
|---|---|---|
| Comba-iplr | $\alpha_t(\boldsymbol{I} - 2\tilde{\beta}_t \boldsymbol{k}_t \boldsymbol{k}_t^\mathsf{T})$ | $(-1, 1)$ |
| Comba-splr | $(\alpha_t - \tilde{\beta}_t \boldsymbol{k}_t \boldsymbol{k}_t^\mathsf{T})$ | $(-1^{\sim 0}, 1)$ |

Moreover, recent efforts [12, 55, 11, 69] aim to distill Transformers into recurrent structures to leverage prior knowledge of large-scale pretrained Transformers, where a key design principle is to minimize additional parameters. To this end, Comba adopts an SPLR structure that only introduces a data-dependent scalar, achieving superior empirical performance compared to RWKV-7 (as implemented in FLA [109]), along with a $2\times$ pretraining acceleration (the memory usage is only half). Recent works [83, 36] have also improved IPLR structures by extending their eigenvalues into the negative domain, e.g., $\boldsymbol{I} - 2\beta_t \boldsymbol{k}_t \boldsymbol{k}_t^\mathsf{T}$, to enhance the model's state-tracking capability. Notably, SPLR naturally admits negative eigenvalues. For fair comparison, in Table 4, we introduce a variant named Comba-iplr. However, such models with global state decay tend to overfit, and the SPLR structure remains optimal and is more structurally aligned with control theory.

**Output Correction**    Comba introduces additional output feedback, from an optimization perspective, this is equivalent to incorporating a similarity optimization objective $\langle \boldsymbol{q}_t, d\boldsymbol{k}_t \rangle$ with factor $d$. In a neural memory perspective, $\boldsymbol{k}$ ensures that memory $\boldsymbol{v}$ is stored as clearly as possible, enabling precise querying by $\boldsymbol{q}$. This optimization objective directly facilitates this process, and significantly reduces the model's perplexity, thereby enhancing performance (§4.2). Empirically, initializing $d$

to 0.02 improves performance for some smaller models (e.g., 340M), enabling gradual learning of the similarity between $q$ and $k$. For larger models (e.g., 1.3B, 2.7B), initializing $d$ to 1 leads to the greatest performance[2]. While prior research has largely focused on improving gating mechanisms or optimizing state updates via key-value operations, few models have explicitly modified the output pathway (i.e., the query @ State). We think there appears to be a potential connection between Comba and MesaNet [96], as MesaNet also performs query correction in the output stage by solving a closed-form recursive least squares problem [10], resulting in $q_t = (H_t + \Lambda)^{-1} q_t$. Notably, the correction in Comba can be implemented with a single line of code $q_t - dk_t$ and is applicable to nearly all RNN variants. In App. A.2, we also provide an alternative explanation, namely that $d$ is equivalent to the residual matrix $D$ in Mamba [38].

Table 5: Various initialization examples and numerical range of the existing recurrent model gates.

| Model | Forget Gate $\alpha_t$ | Range | Input Gate $\beta_t$ | Range |
|---|---|---|---|---|
| GLA [107] | $\mathrm{sigmoid}(W_1 W_2 x_t)^{\frac{1}{\tau}} \mathbf{1}^\intercal$ | $(0, 1)$ | N/A | 1 |
| Mamba2 [26] | $\exp\left(-a\,\mathrm{softplus}\left(W_\alpha x_t + c\right)\right)$ | $\sim 1$ | $\mathrm{softplus}\left(W_\alpha x_t + c\right)$ | $\sim 0$ |
| MetaLA [22] | $\mathrm{sigmoid}(W_\alpha x_t)\mathbf{1}^\intercal$ | $(0, 1)$ | $\mathbf{1} - \mathrm{sigmoid}(W_\alpha x_t)$ | $(0, 1)$ |
| Comba (**ours**) | $\exp\left(-a\,\mathrm{softplus}\left(W_\alpha x_t + c\right)\right)$ | $\sim 1$ | $\mathrm{sigmoid}(W_\beta x_t)$ | $\tilde{\beta}_t < \beta_t \in (0, 1)$ |

## 3.1 Forcing Forgetting for Long-range Modeling

Extensive work [61, 92, 76] has shown that positional encoding is crucial for language models to generalize from short pretraining contexts to unconstrained inference lengths. While gated linear recurrent structures themselves can be viewed as the continuous accumulation of bias [92], enabling these models to extrapolate effectively by learning how to forget. Recent studies [20, 110, 9] suggest that pretraining lengths (e.g., 2K or 4K tokens) are insufficient to fill the model's state capacity, thus making the forget gate close to 1, forcing the model to learn how to forget. As shown in Table 5, we evaluate representative gating initialization methods. For models like GLA [107], the forget gate is initialized to $(0, 1)$ and retains all incremental memories. For MetaLA [22], HGRN2 [79], and GSA [112], the sum of the forget and input gates is constrained to 1, achieving a relative balance in system memory capacity. Experimental results in §4.2 indicate that breaking this balance is necessary. To this end, Comba follows Mamba2's design by constraining the forget gate $\alpha_t$ to close to 1, while separately initializing the input gate $\beta_t$ to $(0, 1)$ [108]. Additionally, we set the strength of state feedback correction is $\tilde{\beta}_t = b \odot \beta_t$, where $b$ is computed by $\mathrm{Sigmoid}$ function to be constrained in interval $(0, 1)$ to ensure that the feedback strength is weaker than the incremental information.

## 3.2 Comba with Chunk-wise Parallel

Based on Eq. 5, Comba can be implemented recursively, enabling constant memory usage and $\mathcal{O}(1)$ time complexity during the inference stage, where the Python-style pseudocode is provided in App. B.1. However, the naive recursive implementation in PyTorch lacks sufficient matrix multiplication and thread parallelism [25], resulting in unacceptable overhead to pretraining. Referring to DeltaNet [105], we optimize Comba through chunk-wise parallelism. App. B.3 presents an alternative form to fuse feedback decay factor $b$ into $k$ as $p = bk$ for a flexible invocation.

In the following illustration, $\square_{[t]} \in \mathbb{R}^{C \times d}$ for $\square \in \{Q, K, V, O, U, W\}$ defines the chunkwise matrices that stack the $q_t, k_t, v_t, o_t, u_t, w_t$ vectors. Additionally, we set $\square_{[t]}^{1:r} = \prod_{i=tC}^{tC+r} \square_{[t]}^i$, $\mathrm{Diag}(\square_{[t]}^{1 \to r}) = \mathrm{Diag}\{\square_{[t]}^1, \ldots, \square_{[t]}^r\}$, and $\mathcal{A}_{[t]}^{i/j} \in \mathbb{R}^{C \times C}$ is a matrix with element $\alpha_{[t]}^{1:i}/\alpha_{[t]}^{1:j}$.

By partially expanding the recurrence to a chunk-wise formulation for Eq. 5, we have:

$$S_{[t]}^r = S_{[t]}^0 \underbrace{\left( \prod_{i=1}^r \left( \alpha_{[t]}^i - \tilde{\beta}_{[t]}^i k_{[t]}^i k_{[t]}^{i\intercal} \right) \right)}_{:=D_{[t]}^r \text{ ("pseudo" memory decay)}} + \underbrace{\sum_{i=1}^r \left( \beta_{[t]}^i v_{[t]}^i k_{[t]}^{i\intercal} \prod_{j=i+1}^r \left( \alpha_{[t]}^j - \tilde{\beta}_{[t]}^j k_{[t]}^j k_{[t]}^{j\intercal} \right) \right)}_{:=H_{[t]}^r \text{ ("pseudo" Incremental memory)}} \quad (6)$$

---

[2]We find that $d = 1$ yields better performance in most cases, which is the default configuration for Comba.

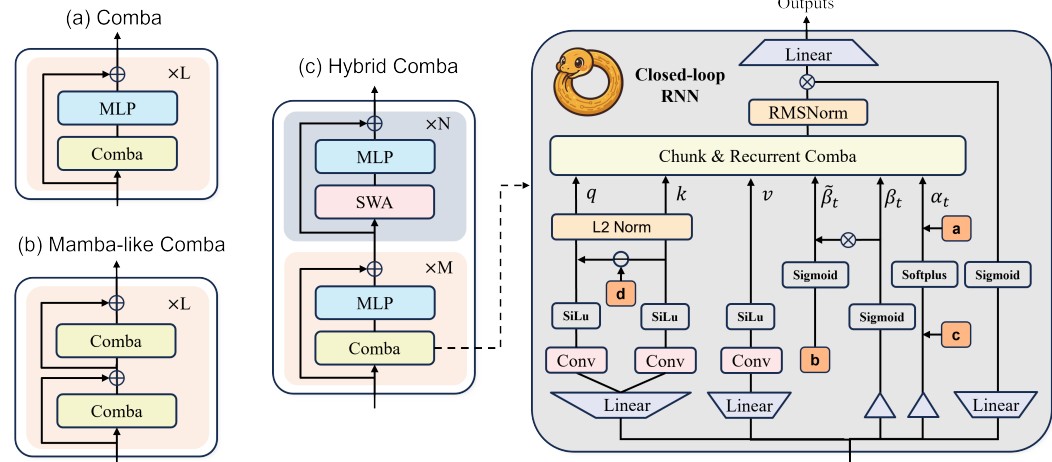

Figure 2: **Comba Families.** The Mamba-like architecture omits MLP layers, uses multi-value attention, and doubles the model depth. For the hybrid model, we incorporate sliding window attention in flexible proportions to boost the model's recall ability. The window size is set to the context length, equivalent to softmax attention.

Eq. 6 involves matrix-matrix products at each time step, i.e., $\boldsymbol{S}_{[t]}^0 @ \boldsymbol{D}_{[t]}^r$, preventing parallelization in the sequence level. Then, we employ the WY representation [13] to eliminate these terms:

$$\boldsymbol{D}_{[t]}^r = \alpha_{[t]}^{1:r} - \sum_{i=1}^{r} \alpha_{[t]}^{i:r} \boldsymbol{w}_{[t]}^i \boldsymbol{k}_{[t]}^{i\mathsf{T}}, \qquad \boldsymbol{w}_{[t]}^r = \tilde{\beta}_{[t]}^r \left( \alpha_{[t]}^{1:r-1} \boldsymbol{k}_{[t]}^r - \sum_{i=1}^{r-1} \boldsymbol{w}_{[t]}^i \left( \alpha_{[t]}^{i:r-1} \boldsymbol{k}_{[t]}^{i\mathsf{T}} \boldsymbol{k}_{[t]}^r \right) \right) \quad (7)$$

$$\boldsymbol{H}_{[t]}^r = \sum_{i=1}^{r} \alpha_{[t]}^{i:r} \boldsymbol{u}_{[t]}^i \boldsymbol{k}_{[t]}^{i\mathsf{T}}, \qquad \boldsymbol{u}_{[t]}^r = \beta_{[t]}^r \boldsymbol{v}_{[t]}^r - \tilde{\beta}_{[t]}^r \sum_{i=1}^{r-1} \boldsymbol{u}_{[t]}^i \left( \alpha_{[t]}^{i:r-1} \boldsymbol{k}_{[t]}^{i\mathsf{T}} \boldsymbol{k}_{[t]}^r \right) \quad (8)$$

To maximize hardware efficiency, we apply the UT transform [51] to Eq. 7-8 to reduce non-matmul FLOPs, which is crucial to enable better hardware utilization during training:

$$\boldsymbol{W}_{[t]} = \boldsymbol{M}_{[t]} \mathrm{Diag}\left( \tilde{\beta}_{[t]}^{1 \to C} \odot \alpha_{[t]}^{0 \to (C-1)} \right) \boldsymbol{K}_{[t]}, \qquad \boldsymbol{U}_{[t]} = \boldsymbol{M}_{[t]} \mathrm{Diag}\left( \beta_{[t]}^{1 \to C} \right) \boldsymbol{V}_{[t]} \quad (9)$$

$$\boldsymbol{M}_{[t]} = \left( \boldsymbol{I} + \mathrm{lower}\left( \mathrm{Diag}\left( \tilde{\beta}_{[t]}^{1 \to C} \right) \left( \mathcal{A}_{[t]}^{(i-1)/j} \odot \boldsymbol{K}_{[t]} \boldsymbol{K}_{[t]}^{\mathsf{T}} \right) \right) \right)^{-1} \quad (10)$$

The inverse of a lower triangular matrix can be efficiently computed through an iterative row-wise approach by forward substitution in Gaussian elimination [37] and maintain data in $\mathrm{float}32$. Notably, Comba computes the inverse matrix once in Eq. 10 (twice in the original Gated-DeltaNet). This is mainly attributed to the introduction of a new form of mathematical induction in deriving the WY representation in Eq. 7, leading to a more concise formulation and resulting in speedup[3].

Finally, we can formulate Eq. 5 in a matrix form to perform chunk-wise parallel training:

$$\boldsymbol{S}_{[t+1]} = \alpha_{[t]}^{1:C} \boldsymbol{S}_{[t]} + \left( \boldsymbol{U}_{[t]} - \boldsymbol{W}_{[t]} \boldsymbol{S}_{[t]}^{\mathsf{T}} \right)^{\mathsf{T}} \mathrm{Diag}\left( \alpha_{[t]}^{i \to C} \right) \boldsymbol{K}_{[t]} \quad (11)$$

$$\boldsymbol{O}_{[t]} = \underbrace{\mathrm{Diag}\left( \alpha_{[t]}^{1 \to C} \right) \tilde{\boldsymbol{Q}}_{[t]} \boldsymbol{S}_{[t]}^{\mathsf{T}}}_{\text{inner chunk}} + \underbrace{\mathrm{Tril}(\tilde{\boldsymbol{Q}}_{[t]} \boldsymbol{K}_{[t]}^{\mathsf{T}} \odot \mathcal{A}_{[t]}^{i/j})}_{\text{intra chunk}} \underbrace{\left( \boldsymbol{U}_{[t]} - \boldsymbol{W}_{[t]} \boldsymbol{S}_{[t]}^{\mathsf{T}} \right)}_{\text{"pseudo"-value term}} \quad (12)$$

where the query matrix $\tilde{\boldsymbol{Q}}_{[t]}$ is also influenced by the feedback control and can be precomputed by: $\tilde{\boldsymbol{Q}}_{[t]} = \boldsymbol{Q}_{[t]} - \mathrm{Diag}(d_{[t]}^{1 \to C}) \boldsymbol{K}_{[t]}$ in chunk-wise at minimal cost.

---

[3]During the forward pass, Comba alleviates the main bottleneck of inverse matrix computation with $\mathcal{O}(d^3)$ in the original Gated-DeltaNet, yielding a 40% speedup. In the backward pass, performance gains diminish due to the reuse of cached inverse matrices $\boldsymbol{M}$. However, in large-scale models where recomputation is required, Comba is expected to offer significant performance advantages. Moreover, this structured modification applies to almost all models with Delta rule. Recently, inspired by Comba, Gated-DeltaNet has also been upgraded to a single inversion in `flash-linear-attention`, resulting in a significant speedup.

### 3.3 Neural Architecture

As shown in Fig. 2, we present the architecture for Comba families, where $\{a, b, c, d\}$ are trainable scalars (experimental results indicate that data dependency is not required). Following prior work [22, 105, 38, 32], we introduce short convolutions to $\boldsymbol{qkv}$ to incorporate token shift to improve the model's retrieval capacity. We also retain feature map operations [98] and utilize the SiLU function to approximate the exponential kernel in the softmax attention. To further stabilize training, we apply L2 normalization to $\boldsymbol{qk}$ and employ a Sigmoid-based gating mechanism. Additionally, we explore a hybrid architecture [89, 105, 80] by integrating Comba layers directly with softmax attention.

## 4 Experiments

**Setting** In this paper, all models are pretrained based on `flash-linear-attention` [109] repository and utilize *NVIDIA A800-80G GPUs*. The 340M Comba pretraining requires *8×10 GPU hours*, while the 1.3B Comba requires *32×48 GPU hours*. We employ the AdamW optimizer [63] with a 3e-4 learning rate, cosine schedule, 0.01 weight decay, and 1.0 gradient clipping. Random seed is 42.

### 4.1 Operator Efficiency Analysis

As shown in Fig. 3, we compared the speeds of various operators in both forward and backward processes. The recurrent Comba in PyTorch [71] incurs significant computational overhead, limiting its scalability for large-scale pretraining. Flash-attention [25] achieves the fastest speed for shorter sequences (e.g., 1024), but its quadratic complexity results in decreasing efficiency as sequence length increases. Among four modern RNN operators, Comba shows nearly 40% speed improvement in the forward process over Gated-DeltaNet due to a more efficient formula structure. GLA suffers from slower operator speed due to the use of diagonal gating matrices. Although RetNet achieves the fastest speed, it falls behind other models in performance due to the lack of data-dependent memory management (Table 6). Overall, Comba shows potential as a foundational framework.

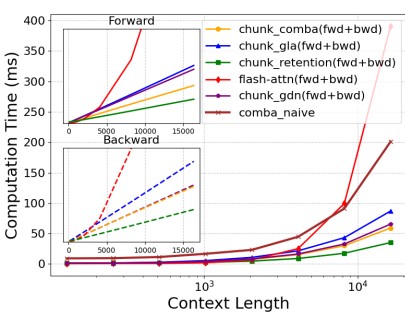

Figure 3: Operator speed evaluated on the `Triton-Testing-Benchmark` [93] (fwd and bwd) in single A800-80G GPU.

### 4.2 Language Modeling & Architecture Ablation

**Commonsense Reasoning Ability** As shown in the left half of Table 6, (i) most recursive models outperform transformers in commonsense reasoning tasks, due to their recursive structure that resembles a chain of thought [100]. (ii) The SPLR structure outperforms both the IPLR and DPLR structures in two model sizes and achieves the highest computational efficiency. (iii) Output correction, i.e., $\langle q, dk \rangle$, significantly reduces perplexity, enhancing memory utilization during question answering and improving performance across various metrics. (iv) We find that the Mamba architecture design is suboptimal. MLP, as a special (non-Hebbian) key-value memory [35, 34], complements the key-value associative memory in the state, which is particularly important for tasks such as inference. (v) Although Mamba's multi-value attention model aids efficient key-value memory storage [59], it sacrifices performance compared to standard multi-head attention. (vi) We discover that the initialization of $d$ should be chosen differently for model scales, e.g., $d = 0.02$ for 340M models and $d = 1$ for 1.3B models, as smaller models are more prone to incorrect gradient descent directions.

Figure 4 shows the training loss curves of various architectures, and Comba, particularly with output correction, demonstrates lower loss and greater expressive power. However, in our experiments, we found that the IPLR structure typically yields lower loss. This may be because (i) as shown in Table 4, the range for the SPLR structure eigenvalues is slightly smaller than that of IPLR, likely due to the special initialization of $\alpha$. (ii) We observe that in visual modeling, the IPLR version of the model often experiences a rapid loss decrease followed by an increase. Combining these findings with the results in Table 6, we speculate that the IPLR structure is prone to overfitting.

**Recall Ability & Hybrid Architecture** As shown in the right half of Table 6, (i) the overall results align with the trends observed in commonsense reasoning tasks. (ii) Recursive models have traditionally struggled with limited recall due to their finite state space, unlike the transformer's unlimited

Table 6: Zero-shot performance of 340M and 1.3B models trained on `SlimPajama` [85] datasets. The commonsense Reasoning task is evaluated by `lm-evaluation-harness` [33] and the recall-intensive task follows `prefix-linear-attention` [3] with 2K input tokens. * Some of the baseline results are from [108] and [29].

| Model & Scale | Lamb.† ppl↓ | Wiki. ppl↓ | ARC_e acc | ARC_c acc_n | Hella. acc_n | Lamb. acc | PIQA acc | Wino. acc | Avg. acc | FDA acc | SWDE acc | SQD. acc | NQ acc | TQA acc | Drop acc | Avg. acc |
|---|---|---|---|---|---|---|---|---|---|---|---|---|---|---|---|---|
| *340M params with 15B training tokens and 0.5M batchsize tokens* | | | | | | | | | | | | | | | | |
| Trans++* | 76.46 | 28.39 | 44.91 | **25.94** | 34.95 | 26.90 | 64.31 | 51.07 | 41.35 | **46.14** | 25.87 | **33.22** | 18.94 | 45.97 | 19.94 | **31.68** |
| GLA | 72.41 | 28.44 | 45.30 | 23.13 | 34.71 | 26.14 | 64.58 | 51.64 | 40.92 | 11.26 | 16.78 | 27.85 | 12.77 | 43.80 | 17.68 | 21.69 |
| Mamba* | 64.75 | 28.39 | 46.30 | 23.60 | 35.40 | 26.72 | 65.00 | 50.10 | 41.80 | 7.14 | 12.96 | 24.35 | 9.47 | 41.84 | 17.11 | 18.81 |
| RWKV7 | 45.00 | 25.74 | 49.03 | 25.09 | 36.63 | 29.01 | 65.45 | 51.54 | 42.79 | 29.34 | **29.15** | 31.81 | 18.21 | **49.17** | 20.56 | 29.71 |
| G-DeltaNet | 45.46 | 26.47 | 46.04 | 23.55 | 37.28 | 29.59 | 66.05 | 50.75 | 42.21 | 20.53 | 23.34 | 28.55 | 14.98 | 44.91 | 16.48 | 24.80 |
| Comba-iplr | **35.37** | 24.31 | 48.15 | 23.04 | 38.01 | **31.71** | 65.83 | 51.62 | 43.06 | 27.98 | 27.66 | 28.92 | 17.96 | 47.75 | 18.35 | 28.10 |
| Comba-splr | 39.91 | **24.15** | **48.56** | 24.32 | 38.18 | 30.98 | **66.73** | 51.41 | **43.36** | 38.51 | 27.61 | 30.07 | 16.38 | 48.60 | **21.22** | 30.40 |
| w/o. $\langle q, dk \rangle$ | 44.91 | 25.49 | 47.94 | 22.78 | 37.93 | 28.96 | 66.43 | 50.67 | 42.45 | 26.33 | 28.02 | 30.03 | 15.64 | 48.93 | 20.36 | 28.21 |
| w/o. $\tilde{\beta} = b \odot \beta$ | 40.17 | 24.56 | 48.37 | 24.36 | 38.02 | 31.18 | 65.53 | 51.36 | 43.13 | 35.66 | 27.70 | 29.31 | 16.72 | 48.10 | 20.84 | 29.72 |
| w/o. $\alpha \sim 1$ | 42.05 | 25.11 | 48.33 | 22.94 | 37.28 | 30.44 | 66.38 | 50.75 | 42.68 | 34.31 | 25.60 | 29.44 | 16.24 | 46.89 | 19.74 | 28.70 |
| w/o. output gate | 40.16 | 24.71 | 48.29 | 23.16 | 37.32 | 29.48 | 66.70 | 50.86 | 42.64 | 31.79 | 23.62 | 29.63 | 18.53 | 48.52 | 20.84 | 28.82 |
| w. Initial d=1 | 39.37 | 24.27 | 47.98 | 22.87 | **38.36** | 30.66 | 66.65 | **53.04** | 43.26 | 30.24 | 26.24 | 28.82 | 15.68 | 48.04 | 18.64 | 27.94 |
| w. mamba-like | 43.20 | 25.45 | 46.04 | 22.95 | 36.96 | 30.76 | 64.36 | 48.78 | 41.64 | 30.64 | 25.31 | 28.44 | 17.33 | 46.68 | 18.82 | 27.87 |
| *1.3B params with 100B training tokens and 1M batchsize tokens* | | | | | | | | | | | | | | | | |
| Trans++* | 19.29 | 17.61 | 55.01 | **28.07** | 49.21 | 40.95 | 70.08 | **56.27** | 49.93 | **44.32** | 32.43 | **42.59** | 24.49 | 58.47 | 21.56 | **37.31** |
| RetNet* | 21.97 | 18.18 | 57.49 | 26.88 | 48.09 | 37.75 | 69.37 | 53.28 | 48.81 | 13.62 | 22.59 | 33.46 | 15.43 | 53.79 | 19.79 | 26.45 |
| GLA* | 19.66 | 17.61 | 55.18 | 27.56 | 48.89 | 40.03 | 69.86 | 53.91 | 49.24 | 27.61 | 30.93 | 35.04 | 22.27 | 56.28 | 19.45 | 31.93 |
| Mamba* | 19.01 | 17.12 | 56.22 | 28.01 | 50.01 | 42.05 | 70.36 | 54.49 | 50.19 | 13.90 | 25.40 | 33.20 | 18.50 | 53.50 | 21.70 | 27.70 |
| G-DeltaNet | 18.80 | 17.14 | 56.82 | 27.39 | 49.77 | 39.94 | 71.76 | 51.78 | 49.58 | 30.25 | 27.65 | 34.06 | 23.22 | 58.23 | 20.36 | 32.29 |
| Comba-iplr | 13.58 | 16.51 | 57.11 | 27.99 | 51.34 | 44.40 | 71.16 | 52.64 | 50.77 | 32.06 | 28.96 | 34.83 | 22.08 | 57.03 | 21.03 | 32.67 |
| Comba-splr | 13.39 | 16.19 | **58.54** | 27.90 | 52.64 | 44.21 | **72.03** | 55.33 | 51.78 | 41.69 | 35.33 | 36.14 | 23.69 | 58.53 | **22.85** | 36.37 |
| w. Initial d=1 | **12.68** | **16.01** | 58.42 | 27.73 | **53.02** | **44.94** | 71.76 | 55.56 | **51.91** | 42.14 | **38.24** | 35.47 | **25.28** | **59.30** | 21.31 | 36.96 |
| w/o. $\langle q, k \rangle$ | 15.64 | 16.94 | 55.39 | 26.02 | 50.30 | 44.65 | 68.82 | 53.12 | 49.72 | 36.97 | 33.55 | 33.96 | 23.66 | 58.12 | 20.65 | 34.49 |

†. For certain reasons, we conduct our evaluation on the lambda-standard dataset rather than on lambda-openai as used in models such as Gated-Deltanet. Consequently, the PPL metric may not be directly comparable with those reported in prior work.

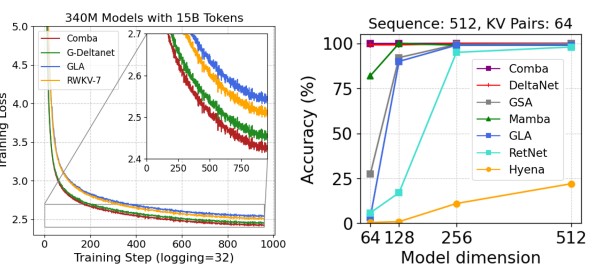

Figure 4: Training loss on 8× A800 GPUs with logging 32.

Figure 5: Results on synthetic MQAR task with settings in [2].

Table 7: Recall-intensive tasks in [3] for hybrid architectures, where the quadratic attention component is implemented using `FlashAttention` [25]. The input length is truncated to 2K.

| | FDA | SWDE | SQD. | NQ | TQA | Drop | Avg. |
|---|---|---|---|---|---|---|---|
| *340M params with 15B tokens* | | | | | | | |
| Trans++ | 46.14 | 25.87 | **33.22** | 18.94 | 45.97 | 19.94 | 31.68 |
| GLA-H | 58.76 | 39.46 | 31.47 | 16.98 | 43.80 | 17.68 | 34.69 |
| GSA-H | **62.13** | **45.36** | 31.17 | **20.62** | 43.78 | 18.78 | **36.97** |
| RWKV7-H | 43.60 | 32.71 | 33.15 | 18.02 | **50.00** | 19.74 | 32.87 |
| GDN-H | 52.13 | 38.80 | 29.90 | 15.20 | 35.55 | 17.73 | 31.55 |
| Comba-H | 60.47 | 37.84 | 31.70 | 16.48 | 47.15 | **20.11** | 35.63 |

KV cache. However, Comba exhibits recall performance very close to that of transformers. (iii) Ablation studies reveal that output correction significantly contributes to Comba's recall performance, optimizing the similarity of $qk$ will enhance the model's ability to retrieve precise memories, thereby improving recall performance. (iv) In Fig. 5, we test Comba's synthetic recall ability on the MQAR task. The Bilinear RNNs, both Gated-DeltaNet and Comba, demonstrate perfect recall ability.

Inspired by previous hybrid architectures [105, 89, 14, 28, 60, 19], we empirically replace the second-to-last layer in every eight layers of Comba with softmax attention to boost the model's recall ability. In Table 7, (i) we find that nearly all hybrid architectures outperform transformers even with a few quadratic layers. (ii) The best overall performance comes from the GSA-H [112], as GSA itself is a type of intra-layer hybrid architecture. Computationally, it can be expressed as two GLA operations followed by a softmax operation to address attention sparsity issues [92], thus improving recall ability. In the future, we plan to combine GSA with Comba for a flexible hybrid.

Table 8: Performance on `LongBench` [4] tasks with 10K length based on `lm-evaluation-harness` [33].

| | Single QA | | | Multi QA | | Summarization | | | Few-shot | | | Code | | |
|---|---|---|---|---|---|---|---|---|---|---|---|---|---|---|---|
| | NQA. | QQA. | MQA. | HQA. | 2WM. | MSQ. | GvR. | QMS. | MNs. | TRE. | TQA. | SAM. | LCC. | RBP. | AVG. |
| Transformer++ | 17.03 | 15.41 | 11.96 | 14.22 | 11.65 | 10.06 | 10.04 | 7.40 | **15.14** | 0.94 | 9.14 | 2.40 | 15.48 | 9.20 | 10.72 |
| RetNet | 25.34 | 24.36 | 21.30 | 24.16 | 23.50 | 21.22 | **14.03** | **11.49** | 12.15 | 7.00 | 22.94 | 6.75 | 17.34 | 19.32 | 17.92 |
| GLA | 26.80 | 25.55 | 23.33 | 24.23 | 26.52 | 23.26 | 11.94 | 7.19 | 9.03 | **9.54** | 27.63 | 5.30 | **20.80** | **21.32** | 18.75 |
| Gated-DeltaNet | 27.62 | 27.55 | 23.34 | 25.19 | 26.63 | 20.26 | 12.33 | 7.24 | 10.51 | 6.83 | 28.42 | 6.07 | 20.37 | 18.12 | 18.61 |
| Comba-splr | **27.73** | **28.56** | **25.78** | **27.49** | **29.55** | **23.34** | 11.61 | 6.20 | 9.47 | 6.58 | **29.63** | **7.11** | 18.04 | 17.07 | **19.16** |
| w/o. $\alpha \sim 1$ | 27.53 | 27.12 | 24.55 | 27.30 | 28.73 | 23.12 | 12.42 | 7.01 | 9.04 | 6.86 | 28.14 | 6.88 | 17.65 | 17.49 | 18.85 |

Table 9: Performance on the ImageNet-1K [27] classification, compared to Vision Mamba [115] (linear), DeiT [94] (quadratic), and Agent Attention [44] (sparse).

| Model | Res. | Params. | FLOPs | Top-1 |
|---|---|---|---|---|
| *DeiT-T* | 224*224 | 5.7(MB) | 1.2(G) | 72.2% |
| Agent-T | 224*224 | 6.0 | 1.2 | 74.9(+2.7) |
| Vim-T | 224*224 | 7.0 | 1.5 | 76.1(+3.9) |
| **Comba-T** | **224*224** | **5.8** | **1.1** | **76.3(+4.1)** |
| *DeiT-S* | 224*224 | 22.1 | 4.6 | 79.8 |
| Agent-S | 224*224 | 22.7 | 4.6 | 80.5(+0.7) |
| Vim-S | 224*224 | 26.0 | 5.1 | 80.3(+0.5) |
| **Comba-S** | **224*224** | **22.6** | **4.4** | **80.5(+0.7)** |

Table 10: Performance on the object tracking datasets such as GOT10k [54] and LaSOT [30], compared to baselines including Vision Mamba (linear), Agent Attention (sparse), and Mixformer [104] (quadratic).

| Model | GOT10k | | | LaSOT | | |
|---|---|---|---|---|---|---|
| | AO | $SR_{0.5}$ | $SR_{0.75}$ | Suc. | N-Pre. | Pre. |
| SA | 0.704 | 0.796 | 0.675 | 0.690 | 0.785 | 0.749 |
| Agent-A | 0.695 | 0.787 | 0.662 | 0.644 | 0.731 | 0.689 |
| mamba vision | 0.700 | 0.789 | 0.673 | 0.677 | 0.771 | 0.730 |
| Comba-splr | 0.715 | 0.804 | 0.686 | 0.693 | **0.789** | 0.751 |
| Comba-iplr | **0.718** | **0.809** | **0.688** | **0.694** | 0.786 | **0.755** |

**Long-context Modeling Ability**   As shown in Table 8, Comba generally outperforms the other architectures, with a significant lead in QA and Few-shot tasks. However, it lags behind Gated-DeltaNet in summarization and code tasks, and the underlying reasons warrant further exploration in the future. Additionally, when the special initialization method in §4 is removed, the model's long-sequence modeling capability decreases, which supports the effectiveness of our approach.

## 4.3   Vision Modeling

**Classification**   As shown in Table 9, Comba achieves SOTA efficiency-accuracy trade-offs across all model scales. For tiny variants, Comba-T improves Top-1 accuracy by 4.1% over DeiT-T with similar parameter count and 8.3% fewer FLOPs. Notably, Comba-T outperforms both Agent-T and Vim-T in accuracy despite requiring fewer computational resources. At small scales, Comba-S matches Agent-S in accuracy while reducing FLOPs by 4.3% and using fewer parameters than Vim-S.

**Object tracking**   To further validate Comba's cross-domain capabilities, we extend experiments to object tracking tasks on GOT-10k and LaSOT datasets. Unlike static image classification, tracking demands efficient temporal modeling and robustness to appearance variations. As shown in Table 10, Comba variants consistently outperform standard attention mechanisms, with the highest AO (0.718) and $SR_{0.75}$ (0.688), exceeding Softmax Attention by 1.4% and 1.3%. Besides, Comba (splr) closely matches Softmax Attention without added computational cost. These results underscore Comba's ability to capture long-range dependencies, such as occlusion recovery and motion continuity.

## 5   Conclusion & Future Work

This paper provides a comprehensive summary of the development of recursive models in efficient sequence modeling methods and highlights the reasons behind the success of the latest generation of Bilinear RNNs. Drawing on closed-loop control theory, we propose Comba, a new architecture that incorporates both state feedback and output correction, based on SPLR state transformations. We also implement a chunk-wise parallel operator using Triton. Extensive experimental results demonstrate the practical advantages of Comba. However, this paper also has several limitations. For instance, due to limited computational resources, the experimental scale was not extended to larger models, such as the 2.7B model (which typically requires *32×120 GPU hours*). Additionally, since models like Titans, Lattice, and MIRAS have not yet been open-sourced, direct comparisons with these models are difficult. In the future, we will focus on addressing the chunk-wise parallel optimization of these models and explore the integration of GSA with the Comba architecture in an elegant hybrid.

## Acknowledgments and Disclosure of Funding

Here, I would like to express my gratitude to those who helped me when I first entered this field, as well as the flash-linear-attention community for their valuable feedback on this paper.

This work is mainly supported by the Guangdong Basic and Applied Basic Research Foundation (No. 2025A1515011994). This work is also supported by the National Natural Science Foundation of China (No. 62402414), Tencent (CCF-Tencent Open Fund, Tencent Rhino-Bird Focused Research Program), Didi (CCF-DiDi GAIA Collaborative Research Funds), Guangzhou Municipal Science and Technology Project (No. 2023A03J0011), Huawei Industrial Funds, and the Guangzhou Industrial Information and Intelligent Key Laboratory Project (No. 2024A03J0628).

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

# Supplementary Material
Improving Bilinear RNNs with Closed-loop Control

## TABLE OF CONTENTS

## A  Additional Background

### A.1  Bilinear Systems

Formally, TTT, Gated-DeltaNet, RWKV-7, and Comba should be classified as bilinear systems [17, 113, 99, 102, 70]. These systems are linear with respect to both $S$ and $K, V$, but due to the interaction between $S$ and $K$, the overall system is nonlinear. Typically considered a special type of nonlinear system, bilinear systems possess more expressive power than linear systems while remaining more controllable than strictly nonlinear systems, such as chaotic systems. They are widely applied in the study of physical systems and biological population dynamics [102, 70].

### A.2  Comba in a State Space Model Perspective

Table 11: Update rules in a control & neural-memory perspective, with feedback $P(\cdot)$ and scalar factor $d$.

| Option | Open-loop Control (Mamba2) | Close-loop Control (Comba) | Gated Delta Rule |
|---|---|---|---|
| *Input / Memorize* | $x_t = \exp(\Delta_t A)x_{t-1} + \Delta_t B_t u_t$ | $x_t = \exp(\Delta_t A)x_{t-1} + \Delta_t B_t u_t^{\text{new}}$ | $S_t = \alpha_t S_{t-1} + \beta_t v_t^{\text{new}} k_t^{\intercal}$ |
| *Feedback / Reflect* | nan | $u_t^{\text{new}} = u_t - P_t(x_{t-1})$ | $v_t^{\text{new}} = v_t - \alpha_t S_{t-1} k_t$ |
| *Output / Recollect* | $o_t = C_t x_t + D u_t$ | $o_t = C_t x_t + D(u_t - P_t(x_t))$ | $o_t = S_t q_t$ |

In Table 11, we provide an explanation of Comba from the perspective of state-space models. Here, $\beta_t$ corresponds to $\Delta_t$, which is an Euler discretization. In state-space models, there is typically a residual term $D u_t$, which we integrate into the residual connection in our framework to omit it.

## B  Operator Implementation Derivation

### B.1  Recurrent Implementation for Comba

We provide the recurrent Comba in Algorithm 1.

```python
def Recurrent_comba(q, k, v, alpha, beta, b, d):
    B, T, H, D = q.shape
    q_new = q - d * k # Output correction
    o, S = torch.zeros_like(v), torch.zeros(b, h, d, d)
    for i in range(T):
        _q, _k, _alpha, _beta = q_new[:, i], k[:, i], alpha[:, i],
    beta[:, i]
```

```
7            _v_new = _beta[..., None] * (v[:, i] - b * (S * _k[..., None])
      .sum(-2))
8          S = _At[..., None] * S + _k.unsqueeze(-1) * _v_new.unsqueeze
      (-2)
9          o[:, i] = torch.einsum('bhd,bhdm->bhm', _q, S)
10     return o
```

Listing 1: Recurrent Comba-pk in Pytorch-like Pseudo-code for Inference

## B.2 WY Representation & UT Transform

Detailed derivation can be found in the Appendix of DeltaNet [108, 105].

## B.3 Comba-SPLR-pk

We also provide an alternative implementation, where $b\boldsymbol{k}$ is integrated into $\boldsymbol{p}$ to make it more close to control theory, these two are equivalent.

```
1  def Recurrent_comba(q, k, v, p, At, dt, D):
2      b, t, h, d = q.shape
3      q_new = q - D[..., None] * p # Output correction
4      o, S = torch.zeros_like(v), torch.zeros(b, h, d, d)
5      for i in range(t):
6          _q, _k, _p, _At, _dt = q_new[:, i], k[:, i], p[:, i], At[:, i
      ], dt[:, i]
7          _v_new = _dt[..., None] * (v[:, i] - (S * _p[..., None]).sum
      (-2))
8          S = _At[..., None] * S + _k.unsqueeze(-1) * _v_new.unsqueeze
      (-2)
9          o[:, i] = torch.einsum('bhd,bhdm->bhm', _q, S)
10     return o
```

Listing 2: Recurrent Comba-pk in Pytorch-like Pseudo-code for Inference

By partially expanding the recurrence for Eq. 5, we have

$$\boldsymbol{S}_{[t]}^r = \boldsymbol{S}_{[t]}^0 \underbrace{\left( \prod_{i=1}^r \left( \alpha_{[t]}^i - \beta_{[t]}^i \boldsymbol{p}_{[t]}^i \boldsymbol{k}_{[t]}^{i\intercal} \right) \right)}_{:=\boldsymbol{D}_{[t]}^r \text{ (``pseudo'' memory decay)}} + \underbrace{\sum_{i=1}^r \left( \beta_{[t]}^i \boldsymbol{v}_{[t]}^i \boldsymbol{k}_{[t]}^{i\intercal} \prod_{j=i+1}^r \left( \alpha_{[t]}^j - \beta_{[t]}^j \boldsymbol{p}_{[t]}^j \boldsymbol{k}_{[t]}^{j\intercal} \right) \right)}_{:=\boldsymbol{H}_{[t]}^r \text{ (``pseudo'' Incremental memory)}}$$

Then, we employ the WY representation [13]:

$$\boldsymbol{D}_{[t]}^r = \alpha_{[t]}^{1:r} - \sum_{i=1}^r \alpha_{[t]}^{i:r} \boldsymbol{w}_{[t]}^i \boldsymbol{k}_{[t]}^{i\intercal} \qquad \boldsymbol{w}_{[t]}^r = \beta_{[t]}^r \left( \alpha_{[t]}^{1:r-1} \boldsymbol{p}_{[t]}^r - \sum_{i=1}^{r-1} \boldsymbol{w}_{[t]}^i \left( \alpha_{[t]}^{i:r-1} \boldsymbol{k}_{[t]}^{i\intercal} \boldsymbol{p}_{[t]}^r \right) \right)$$

$$\boldsymbol{H}_{[t]}^r = \sum_{i=1}^r \alpha_{[t]}^{i:r} \boldsymbol{u}_{[t]}^i \boldsymbol{k}_{[t]}^{i\intercal} \qquad \boldsymbol{u}_{[t]}^r = \beta_{[t]}^r \left( \boldsymbol{v}_{[t]}^r - \sum_{i=1}^{r-1} \boldsymbol{u}_{[t]}^i \left( \alpha_{[t]}^{i:r-1} \boldsymbol{k}_{[t]}^{i\intercal} \boldsymbol{p}_{[t]}^r \right) \right)$$

To maximize hardware efficiency, we apply the UT transform [51] to reduce non-matmul FLOPs, which is crucial to enable better hardware utilization during training.

$$\boldsymbol{W}_{[t]} = \boldsymbol{M}_{[t]} \text{Diag}\left( \beta_{[t]}^{1\to C} \odot \alpha_{[t]}^{0\to(C-1)} \right) \boldsymbol{P}_{[t]}, \qquad \boldsymbol{U}_{[t]} = \boldsymbol{M}_{[t]} \text{Diag}\left( \beta_{[t]}^{1\to C} \right) \boldsymbol{V}_{[t]}$$

$$\boldsymbol{M}_{[t]} = \left( \boldsymbol{I} + \text{lower}\left( \text{Diag}\left( \beta_{[t]}^{1\to C} \right) \left( \mathcal{A}_{[t]}^{(i-1)/j} \odot \boldsymbol{P}_{[t]} \boldsymbol{K}_{[t]}^{\intercal} \right) \right) \right)^{-1}$$

The inverse of a lower triangular matrix can be efficiently computed through an iterative row-wise approach by forward substitution in Gaussian elimination [37] and maintain data in float32.

Then we have the following vector form:

$$\boldsymbol{S}_{[t]}^r = \boldsymbol{S}_{[t]}^0 \boldsymbol{D}_{[t]}^r = \alpha_{[t]}^{1:r} \boldsymbol{S}_{[t]}^0 + \sum_{i=1}^{r} \alpha_{[t]}^{i:r} \left( \boldsymbol{u}_{[t]}^r - \left( \boldsymbol{S}_{[t]}^0 \boldsymbol{w}_{[t]}^i \right) \right) \boldsymbol{k}_{[t]}^{i\mathsf{T}}$$

$$\boldsymbol{o}_{[t]}^r = \boldsymbol{S}_{[t]}^r \tilde{\boldsymbol{q}}_{[t]}^r = \alpha_{[t]}^{1:r} \boldsymbol{S}_{[t]}^0 \tilde{\boldsymbol{q}}_{[t]}^r + \sum_{i=1}^{r} \left( \boldsymbol{u}_{[t]}^r - \left( \boldsymbol{S}_{[t]}^0 \boldsymbol{w}_{[t]}^i \right) \right) \left( \alpha_{[t]}^{i:r} \boldsymbol{k}_{[t]}^{i\mathsf{T}} \tilde{\boldsymbol{q}}_{[t]}^r \right)$$

Equivalently, in matrix form:

$$\boldsymbol{S}_{[t+1]} = \alpha_{[t]}^{1:C} \boldsymbol{S}_{[t]} + \left( \boldsymbol{U}_{[t]} - \boldsymbol{W}_{[t]} \boldsymbol{S}_{[t]}^{\mathsf{T}} \right)^{\mathsf{T}} \operatorname{Diag}\left( \alpha_{[t]}^{i \to C} \right) \boldsymbol{K}_{[t]}$$

$$\boldsymbol{O}_{[t]} = \underbrace{\operatorname{Diag}\left( \alpha_{[t]}^{1 \to C} \right) \tilde{\boldsymbol{Q}}_{[t]} \boldsymbol{S}_{[t]}^{\mathsf{T}}}_{\text{inner chunk}} + \underbrace{\operatorname{Tril}(\tilde{\boldsymbol{Q}}_{[t]} \boldsymbol{K}_{[t]}^{\mathsf{T}} \odot \mathcal{A}_{[t]}^{i/j})}_{\text{intra chunk}} \underbrace{\left( \boldsymbol{U}_{[t]} - \boldsymbol{W}_{[t]} \boldsymbol{S}_{[t]}^{\mathsf{T}} \right)}_{\text{``pseudo''-value term}}$$

where the query matrix $\tilde{\boldsymbol{Q}}$ is also influenced by the closed-loop control and can be precomputed by: $\tilde{\boldsymbol{Q}} = \boldsymbol{Q} - \operatorname{Diag}(d_{[t]}^{1 \to C}) \boldsymbol{P}_{[t]}$.

