# OpenReview forum: "Improving Bilinear RNN with Closed-loop Control"
_NeurIPS.cc/2025/Conference — NeurIPS 2025 spotlight_

### Official Review · Reviewer_npnS · 2025-06-18

**Clarity:** 3
**Significance:** 3
**Originality:** 3
**Rating:** 5
**Confidence:** 4

**Summary:**

## Summary

In this article, the authors provide a detailed literature review on recursive models for efficient processing of sequential data. This synthesis highlights the strengths and weaknesses of each model and thus allows them to propose a novel RNN architecture that aims to incorporate dependencies on the previous state and the output while remaining efficient during inference. In addition to the detailed literature summary, the authors support their new architecture with experiments that show that the new model is competitive with the state of the art, and in a number of tasks outperforms the state of the art.

**Questions:**

## Questions
Anglais

Do you have any idea if it is theoretically possible to show the expressiveness of your model and if possible place it in the ecosystem of existing models?

**Ethical Concerns:**

["NO or VERY MINOR ethics concerns only"]

**Limitations:**

Due to the unavailability of these state-of-the-art models, the authors were unable to compare their models to the latest state-of-the-art models. Due to infrastructure limitations, they were unable to conduct very large-scale experiments.

**Quality:**

4

**Strengths And Weaknesses:**

## Strengths
- The article is very well written, well -illustrated and clear.
- The authors have a deep knowledge of literature and know exactly how their work is positioned.
- The methodology of numérical experiments is rigorous.

## Weaknesses
- There is a lack of theory to support the claims. Validations are purely empirical.
- As the authors themselves said, due to infrastructure limitations, they were unable to conduct very large-scale experiments.

---

> ### Author Rebuttal · Authors · 2025-07-30
>
> We sincerely thank you for your appreciation of our work. Below, we provide further clarification regarding the two minor concerns you raised. Lastly, we would also like to share our perspective on Comba's positioning within deep learning architectures and its future directions.
>
> # Q1: The lack of theory to support the claims.
>
> Thank you for raising this point. We think that the development of modern deep learning architectures has been largely driven by empirical insights and intuition. For example, the attention mechanism in the original Transformer was introduced empirically, with theoretical analyses only emerging in subsequent years. Similarly, the early development of structured state-space models (SSMs) relied on theory-driven parameter initialization, such as using HiPPO-based matrices. However, these were eventually outperformed by models like Mamba, which are entirely data-driven.
>
> We acknowledge that theory plays a crucial role in understanding **why** a model or mechanism works. Nonetheless, practical deployment often involves complexities that exceed the reach of current theoretical tools. In the revised version of the paper, we will provide a theoretical characterization of Comba's expressiveness. In particular, from the perspective of computational circuit complexity [1,2,3,4], we show that Comba belongs to the **NC¹** class, which strictly exceeds the **TC⁰** class associated with Transformer-based architectures.
>
>
> # Q2: Very large-scale experiments.
>
> Since Comba has open-sourced in Flash-Linear-Attention library, it has attracted growing interest from the community. We are now collaborating with a technology company to conduct large-scale experiments to further evaluate Comba's performance, and we warmly welcome you to try out Comba.
>
>
> # Q3: The Position of Comba in the ecosystem of deep learning architectures.
>
> | Expressive/Efficient | `O(T)` | `O(Tlog T)` | `O(T^2)` |
> |------------|--------|------------|--------|
> | P      | LSTM |       --     |    --    |
> | AC¹     |    --      |      --      |    --    |
> | NC¹     | Comba, DeltaNet, RWKV7 | Loglinear Comba | Deltaformer, 2‑Simplicial Attention |
> | TC⁰     | Mamba, MesaNet, RWKV6   | Loglinear Mamba | Transformer |
> |
>
> We believe that deep learning architecture is undergoing a fundamental shift. As the limitations of the Transformer architecture become more widely recognized, a new wave of alternative designs has emerged. When evaluating the quality of a model, two key aspects are typically considered: **expressive power** and **computational efficiency**. As shown in the table, existing architectures can be roughly categorized by their time complexity into **linear**, **log-linear**, and **quadratic** classes (represented on the horizontal axis).
>
> Regarding expressive power, **circuit complexity** has become a common theoretical framework, where one studies the ability of a circuit to simulate the behavior of a neural network. The standard hierarchy of circuit complexity classes is as follows:
>
> $$
> \mathsf{P} > \mathsf{TC}^n > \mathsf{AC}^n > \mathsf{NC}^n > \mathsf{TC}^{n-1} > \mathsf{AC}^{n-1} > \mathsf{NC}^{n-1} > \cdots > \mathsf{TC}^0 > \mathsf{AC}^0 > \mathsf{NC}^0
> $$
>
> Here, the superscript $n$ denotes the logarithmic depth of the computation paths, while $\mathsf{NC}$, $\mathsf{AC}$, and $\mathsf{TC}$ refer to distinct classes of circuit complexity. $\mathsf{P}$ denotes the class of problems solvable in polynomial time, which includes the vast majority of real-world tasks.
>
> Recent studies have shown that models with limited depth and finite-precision arithmetic—such as Transformers and Mamba [5]—are provably within the $\mathsf{TC}^0$ class. In contrast, models like **Comba**, **RWKV-7** [4], and **DeltaNet** [6] can theoretically achieve $\mathsf{NC}^1$-level complexity, which provides strictly greater expressive power. Among these, **Comba** is the most expressive due to its dual-stage feedback design. Although **LSTM** architectures apply nonlinear activation functions to the hidden state, thereby achieving expressive power equivalent to the complexity class **P**, they inherently lack the ability to perform parallel computation across time steps. This sequential dependency renders them unsuitable for large-scale language model pretraining, where parallelism is crucial for computational efficiency. As a result, despite their theoretical expressiveness, LSTMs have limited practical relevance in modern large-scale sequence modeling tasks.
>
>
>
> Moreover, recent efforts in optimizing quadratic-time models have also elevated their theoretical capacity to the $\mathsf{NC}^1$ class, such as Deltaformer [1]. We believe that a promising future direction is to maintain the **efficiency of RNN-style architectures** while further advancing their **expressiveness toward the $\mathsf{AC}^1$ class**.
>
> > We hope our responses address your concerns, and we warmly welcome continued discussion. Thank you again, and we wish you all the best.
>
> [1] Understanding Transformer from the Perspective of Associative Memory
>
> [2] The Illusion of State in State-Space Models
>
> [3] The Parallelism Tradeoff: Limitations of Log-Precision Transformers
>
> [4] Rwkv-7" goose" with expressive dynamic state evolution
>
> [5] Mamba: Linear-time sequence modeling with selective state spaces
>
> [6] Parallelizing Linear Transformers with the Delta Rule over Sequence Length

---

### Official Review · Reviewer_ScfA · 2025-07-01

**Clarity:** 2
**Significance:** 3
**Originality:** 3
**Rating:** 5
**Confidence:** 2

**Summary:**

This paper introduces a new nonlinear recurrent neural network (RNN) model called Comba, which is inspired by control theory. The authors review recent nonlinear RNNs that use Delta learning rules to improve memory control. Unlike earlier models, Comba uses a scalar-plus-low-rank (SPLR) state update and adds feedback both to the state and output, forming a closed-loop system. It is also designed to be hardware-efficient by supporting chunk-wise parallel computation. The authors test Comba on various language and vision tasks, showing it has better performance and faster training compared to other models. They also explore hybrid designs and provide detailed experiments and analysis.

**Questions:**

1. I find the discussion on whether models like Mamba2 are considered open-loop or closed-loop somewhat confusing. From a general perspective, most RNNs, including Mamba2, are inherently “closed-loop” in the sense that the current state S_t depends on the previous state S_{t-1}, which serves as a form of feedback. This contrasts with autoregressive models like Transformers (with KV cache), which are more “open-loop” because they do not incorporate such recurrent state feedback in the same way. However, in Section 3, the paper seems to classify Mamba2 as an open-loop control system. Does this mean that the term “closed-loop” in this work specifically refers to the inclusion of negative feedback mechanisms, such as corrective terms that subtract projections from previous states (e.g., v_t - P_t(S_{t-1}))? If so, it would be helpful to explicitly distinguish this interpretation from the broader control-theoretic or RNN-centric notions of feedback.

2.The experimental results convincingly demonstrate that the proposed model achieves strong performance. However, I still find it unclear whether the negative feedback mechanism is truly the key contributing factor. Could the authors provide more explanation, either theoretical or empirical, on why negative feedback helps in this setting? For instance, does it improve stability, memory precision, or long-range dependency handling in a measurable way? A deeper discussion or ablation focusing specifically on the feedback terms would be very helpful.

**Ethical Concerns:**

["NO or VERY MINOR ethics concerns only"]

**Final Justification:**

I appreciate the authors’ response and will keep my original score.

**Limitations:**

yes

**Paper Formatting Concerns:**

I dont find any major formatting issues.

**Quality:**

3

**Strengths And Weaknesses:**

Strengths:
1. The paper introduces Comba, a new nonlinear RNN architecture that incorporates closed-loop control by combining state feedback and output correction.
2. The authors provide an optimized chunk-wise parallel implementation in Triton, achieving a good speed improvement in forward pass. This makes the method more practical for large-scale training.
3. Comba is rigorously evaluated across multiple benchmarks, including language modeling, commonsense reasoning, long-context understanding (LongBench), and etc.. Results consistently show competitive or superior performance.
4. The work nicely bridges the gap between SSMs, gated linear attention, and Delta-rule-based updates. It also provides a unified perspective on “Nonlinear RNNs” and positions Comba as a foundation model candidate.
5. The authors conduct comprehensive ablations on architecture variants (e.g., SPLR vs. IPLR), feedback mechanisms, and output correction terms, giving credibility to their design choices.

Weaknesses:
1. The paper is not easy to follow for readers who are not already familiar with recent state-space models like Mamba, or with LLM-centric efficient RNNs.
2. The term “Nonlinear RNN” in the title may be confusing for researchers coming from classical RNN/statistical learning backgrounds, where RNNs are already nonlinear in most cases by design. It might benefit from a subtitle or clarification.
3. The notation is dense and not well-initialized early in the paper. Symbols  are used heavily but not clearly introduced or contextualized for new readers. This affects the clarity of theoretical sections.

---

> ### Author Rebuttal · Authors · 2025-07-30
>
> We sincerely thank you for your appreciation of our work. Below, we provide further clarification regarding the concerns you raised.
>
> # W1: Hard to understand for RNN beginners.
> Thank you for bringing this to our attention. In fact, during the writing of this paper, we made a concerted effort to outline the development of modern RNN architectures as thoroughly as possible in the Preliminaries section. We also sought feedback from several non-expert readers to ensure accessibility. However, linear-time sequence modeling architectures often rely on substantial prior knowledge, such as orthogonal polynomial projections, neurobiology, and algebra, which poses a significant challenge for beginners to grasp quickly. We sincerely appreciate your suggestion, and in the revised version, we will include an extended Related Work section in the appendix to help readers better understand the evolution of modern RNNs.
>
> # W2: The concept of Nonlinear RNNs.
>
> Thank you for your valuable feedback. We also became aware of this issue during the review process, and have accordingly updated the term `Nonlinear RNN` to `Bilinear RNN` in the revised version of the paper. This change only involves replacing the terminology, and does not affect any other content in the manuscript. (Due to NeurIPS rebuttal policies, we are unable to update the submitted version.)
>
> We believe that `Bilinear RNN` is a more appropriate designation. Comba is neither strictly nonlinear in the way that LSTMs are, where nonlinearity is applied directly to the hidden state, nor is it merely a linear key-value memory mechanism as in linear attention or state space models. Instead, Comba introduces a product between the key $\mathbf{k}$ and the hidden state $\mathbf{S}$, which corresponds to a `Bilinear System` in control theory [1, 2]. Such systems are widely used for modeling biological populations and physical dynamics, and are also regarded as one of the most expressive controllable system classes.
>
> [1] Bilinear systems: An appealing class of ”nearly linear” systems in theory and applications.
>
> [2] Expectation-maximization algorithm for bilinear state-space models with time-varying delays under non-gaussian noise.
>
> # W3: The notation is unclear.
> Thank you for highlighting this important point. The notation used in Comba is aligned with many prior works on linear time-series modeling [4,5,6,7]. Additionally, we provided a clarification of the symbols in the footnote at the bottom of Table 1.
>
> We apologize for any confusion caused by the lack of explicit notation definitions. In the revised version of the paper, we will include a dedicated **Notation** section before the **Preliminaries** to explain the meaning of symbols used throughout the paper, making the content more accessible for beginners.
>
> [4] Gated slot attention for efficient linear-time sequence modeling
>
> [5] Gated delta networks: Improving mamba2 with delta rule
>
> [6] Gated linear attention transformers with hardware-efficient training
>
> [7] Understanding Transformer from the Perspective of Associative Memory
>
> # Q1: The Concept of Closed-loop System in Comba.
>
> Thank you for your suggestion. In the revised version of the paper, we will clarify the concept of **closed-loop systems**, and confirm that the term "feedback" in Comba refers specifically to **negative feedback**.
>
> In this work, we adopt the fundamental definition of closed-loop systems as a design intuition. As stated on Wikipedia [8]:
>
> > *A closed-loop controller is a control loop which incorporates feedback. A closed-loop controller uses feedback to control states or outputs of a dynamical system.*
>
> Following this definition, consider the standard RNN update: $S_t = S_{t-1} + v_t k_t^T$. Here, $S_{t-1}$ passively receives the update and does not influence the update behavior through feedback. Therefore, models like **Mamba2** can be viewed as **open-loop systems**.
>
> In contrast, **Comba** introduces two explicit forms of feedback:
>
> - **State correction:** $S_t = S_{t-1} (\alpha_t - k_t k_t^T) + v_t k_t^T$
>
> - **Output correction:** $o_t = S_t q_t + d(v_t - S_t k_t) = S_t(q_t - dk_t)$ (Residual connection $d * v_t$ is omitted, as is also done in Mamba2.)
>
> These mechanisms correspond directly to the two parts of the closed-loop definition: feedback for both **states** and **outputs**. In comparison, models like **RWKV-7** and **DeltaNet** only implement feedback at the **state** level and thus cannot be considered end-to-end closed-loop systems. Empirically, we observe that introducing output correction significantly boosts model performance. Notably, this mechanism can be implemented with a **single line of code** `q_t - d*k_t`, and can be easily integrated into many RNN variants. For example, adding this to Gated-DeltaNet and GLA yielded consistent improvements:
>
> - Gated-DeltaNet (1.3B/100B): loss reduced from 2.213 → 2.175
>
> - GLA (1.3B/100B): loss reduced from 2.232 → 2.204
>
> To avoid potential ambiguity, we will explicitly address these clarifications in the revised manuscript.
>
>
> [8] Wikipedia: Closed-loop controller
>
>
>
>
> # Q2: How does feedback improve performance?
>
> Comba incorporates two forms of feedback (as shown in Equation 5): **state correction**, represented by $(\alpha_t - \mathbf{k}_t\mathbf{k}_t^{T})$, and **output correction**, represented by $(\mathbf{q}_t - d\mathbf{k}_t)$. We provide a detailed explanation of both mechanisms below. The effectiveness of **negative feedback** has been well established in the control systems literature. In this work, our main contribution is to reinterpret and implement such feedback mechanisms within the context of **deep learning RNN architectures**:
>
> ## State Correction.
>
> (a). Comba can be reformulated in the following equivalent form: $S_t = \alpha_t S_{t-1} + \beta_t(v_t -  S_{t-1} k_t)k_t^{T}$
>
> Here, the term $(v_t - S_{t-1} k_t)$ can be interpreted as a **new value**, which replaces the original $v_t$ in the outer product update. The intuition behind this formulation is as follows:
>
> In modern RNN architectures, the hidden state $S$ is typically updated via an outer product between the key and value vectors. When querying the memory with $k_t$, the model retrieves the most relevant historical value using $S_{t-1}k_t$. Since the state matrix $S$ has a fixed size, its memory capacity is inherently limited. By using the **difference** between the new value and the retrieved one, i.e., $(v_t - S_{t-1}k_t)$, the model effectively learns **only the information that was not already stored**. This can be seen as a form of memory-efficient update: the smaller the difference, the more redundant the current value is with respect to past knowledge. Thus, this correction mechanism equips the model with **stronger memory management**, promoting efficient utilization of limited hidden state capacity.
>
>
> (b). The state correction matrix $(\alpha_t - b \cdot k_tk_t^{T})$ can be interpreted as a mirror transformation. As illustrated in Figure 1, its geometric meaning is to reflect each column of the memory matrix $\mathbf{S}_{t-1}$ across a hyperplane whose normal is defined by the vector $\mathbf{k}_t$. Intuitively, whenever the model encounters new information, this transformation—scaled by $\beta_t$—suppresses the component of the memory update that lies in the direction of $\mathbf{k}_t$, preserving only the orthogonal component, i.e., the part that represents *novel* or *previously unseen* knowledge.
>
>
> (c). Comba can be rewritten in an explicit Stochastic Gradient Descent (SGD) form:
>
> $$
> S_t = \alpha_t S_{t-1} - \beta_t \nabla_S\left \|\| v_t - S_{t-1} k_t \right \|\|^2
> $$
>
> This formulation implies that the model not only updates its parameters during training, but also continuously adapts its memory state (as a special parameter) during inference. This behavior reflects the core idea of **negative feedback**: the system adjusts itself in real-time to better align with external inputs, enabling continual adaptation to changing environments.
>
>
>
> ## Output Correction
>
> While prior RNN research has largely focused on improving gating mechanisms or optimizing state updates via key-value operations, few models have explicitly modified the output pathway (i.e., the query @ State). We think there appears to be a potential connection between Comba and MesaNet [9] (a recent SOTA model from Google), as MesaNet also performs query correction in the output stage by solving a closed-form recursive least squares problem, resulting in $q_t=(H_t + \Lambda)^{-1}q_t$. Notably, the correction in Comba is much simpler **with a single line of code** `q_t - d*k_t`, and is applicable to nearly all RNN variants.
>
> [9] MesaNet: Sequence Modeling by Locally Optimal Test-Time Training
>
> > We hope our responses address your concerns, and we warmly welcome continued discussion. Thank you again, and we wish you all the best.

---

### Official Review · Reviewer_fcuH · 2025-07-01

**Clarity:** 2
**Significance:** 3
**Originality:** 3
**Rating:** 5
**Confidence:** 2

**Summary:**

This paper presents Comba, a nonlinear RNN model with SPLR state transition and output correction to improve the model performance. Using chunkwise parallelization, Comba can be efficiently trained, with 40% speedup compared to Gated DeltaNet. The effectiveness of two proposed techniques is confirmed by experimental results on language and vision tasks.

**Questions:**

Questions:

1. What is the required memory size for different sequence lengths during training? How does it compare with the Transformer and other modern RNN variants?

2. Eq. 3 seems to only hold for $\alpha=1$. Is this intended?

3. In Table 2, the input dimension of Mamba2 is said to be $2D/H$. Should it be $2D$?

4. Appendix A.1 is empty. Is this intended?

5. Based on Table 1, the difference between the state transition equations of Gated DeltaNet and Comba seems to be the parametrization of $\alpha_t$ and $\beta_t$. In this case, why is Comba sufficiently faster than Gated DeltaNet? Could you apply the method in Sec. 3.2 to Gated DeltaNet similarly so that only one matrix inverse is computed?

**Ethical Concerns:**

["NO or VERY MINOR ethics concerns only"]

**Final Justification:**

This paper introduces Comba, a bilinear RNN model with state and output correction to improve the model performance upon other recent RNN models, such as Mamba 2 and Gated-DeltaNet. Using chunkwise parallelization, Comba can be efficiently trained, with 40% speedup compared to Gated DeltaNet. The effectiveness of the proposed techniques is confirmed by experimental results on language and vision tasks.

During the rebuttal period, the authors have addressed my initial concerns and questions, such as the usage of the term ‘closed-loop control’, the initial value of $d$, and the training memory costs. Given that the topic is highly relevant to the community and the paper provides an effective solution with convincing results, I will raise my score by 1.

**Limitations:**

Yes: the authors have adequately addressed the limitations and potential negative societal impact of their work

**Paper Formatting Concerns:**

No concerns

**Quality:**

3

**Strengths And Weaknesses:**

Strengths:

1. Sec. 2 and Table. 1 provide an informative overview of the development of linear and nonlinear RNNs and the motivations behind these designs.

2. The effectiveness of SPLR and output correction are verified through extensive ablation studies; several variants of Comba are explored to justify the design decisions.

3. Open access to the code is provided.

Weaknesses:

1. It is claimed in the title and Sec. 3 that the design of Comba is based on closed-loop control. However, it is quite difficult to identify the key components (i.e., controller, plant, error detector, feedback element, etc.) of a closed-loop control system from the equations in Table 3 and Eq. 5. In addition, it is difficult to see how closed-loop control helps to improve model performance, besides the general claim that negative feedback improves system adaptability (Line 148). I believe providing a block diagram (as commonly seen in the control literature) for each column of Table 3 would provide better insight to Comba.

2. The output correction seems to be a feed-forward instead of a feedback path, because the output $\boldsymbol{o}_t$ is still a linear transforma of the state $\boldsymbol{S}_t$, just replacing the projection vector from $\boldsymbol{q}_t$ to $\boldsymbol{q}_t-d\boldsymbol{k}_t$, and both $\boldsymbol{q}_t$ & $\boldsymbol{k}_t$ depend on the instantaneous input (maybe I misunderstood what ‘feedback’ exactly means in Comba – see W.1). In this case, it might be incorrect to claim that TTT/DeltaNet/RWKV-7 are not strictly ‘closed-loop systems’ (Line 155) and that Comba has two-stage feedback (Line 149).

3. The initial value of $d$ (for output correction) seems to affect the model performance substantially, and the optimal choice of the initial value seems to heavily depend on the model size (50x difference in $d$ for 3x difference in model size). Investigating the model performance for other values of $d$ (between 0.02 and 1) would help practitioners in choosing this hyperparameter.
\end{enumerate}

---

> ### Author Rebuttal · Authors · 2025-07-30
>
> We sincerely thank you for your rigorous and thoughtful feedback. We will begin by summarizing your concern, followed by a detailed response. Comba has already been open-sourced and is currently under evaluation by the broader LLM research community.
>
> # Q1: Training Cost.
> Thank you for raising this point. We believe deep learning architectures are undergoing a fundamental shift, with increasing attention to Transformer alternatives. To be viable, new architectures must match Transformers in pretraining efficiency.
>
> Comba achieves this through carefully designed kernels while remaining fully aligned with the standard Transformer parameterization—relying primarily on `q`, `k`, `v`, and output projections (**lines 173–176**). The additional terms ($\alpha_t$, $\beta_t$, $b$, $d$) are head-wise scalars with negligible memory overhead.
>
> As detailed in our response to **Question 5**, Comba reduces memory usage by avoiding half of the matrix inversions required in Gated-DeltaNet. Besides, Comba's kernel minimizes intermediate storage, and compared to RWKV7, Comba achieves nearly **100% memory reduction** and up to **2× throughput** improvement. Below we show the GPU memory usage for a 340M model on a single A800-80GB GPU (totally 8×A800 GPUs with batch size 32):
> | A800 | FlashAttentionV2 | Comba |Gated-DeltaNet |RWKV7 |
> |-|-|-|-|-|
> |80GB| 62.26G  | 68.58G |  72.34G  | 132.52G (Out of memory)
>
> # Q2: The $\alpha$ in equation 3.
> In fact, DeltaNet has two versions. The first version was published at NeurIPS 2024 [1], and the second version, Gated-DeltaNet, was published at ICLR 2025 [2]. In line 103, we added a parenthesis around `Gated` to indicate that Equation 3 is explained using the basic DeltaNet, and therefore does not include the parameter $\alpha$.
>
> In fact, the gating formulation in Gated-DeltaNet is incorrect, as it cannot be explicitly written in the standard Stochastic Gradient Descent (SGD) form, while Comba adopts the correct formulation. This issue has also been pointed out in several recent blogs (in accordance with NeurIPS rebuttal guidelines, we cannot include external links):
>
> **Gated-DeltaNet**: $S_t = \alpha_t S_{t-1} - \beta_t \nabla_S\|\| \frac{1}{\alpha_t}v_t-\alpha_t S_{t-1}k_t \|\|^2$
>
> **Comba**: $S_t = \alpha_tS_{t-1} - \beta_t \nabla_S\|\| v_t-S_{t-1}k_t \|\|^2$
>
> [1] Parallelizing Linear Transformers with the Delta Rule over Sequence Length
>
> [2] Gated delta networks: Improving mamba2 with delta rule
>
>
> # Q3: The Dimension for Mamba2.
> In our paper, the dimensionality related to Mamba2 ($\mathbf{u} \in \mathbb{R}^{2D/H}$) is correct. Because the State size in the table refers to the total dimension within one layer of the model, and the input value refers to the head dimension. Therefore, the dimension of the input value in Mamba2 carries a denominator of $H$, while the State size eliminates $H$.
>
> We apologize for the confusion caused by our unclear notation. In the revised version of the paper, we will update `input value` to `input value per head` and `State size` to `All State Size`. A discussion on the dimensionality of Mamba2 can also be found in Appendix A.1 of another paper [3].
>
> [3] Stuffed Mamba: State Collapse and State Capacity of RNN-Based Long-Context Modeling
>
> # Q4: The Appendix A.1.
> We apologize for this oversight. In fact, the paper does not contain an Appendix A.1. This is because we have already provided a comprehensive review of the development of linear RNNs in the Introduction and Section 2. The heading for Appendix A.1 was mistakenly left in the submission version of the paper, and we will remove it in the revised version.
>
> # Q5: Why is Comba faster than Gated-DeltaNet
>
> We believe the primary contribution of Comba lies in its direct acceleration of DeltaNet-style operators. Specifically, Comba achieves up to a **46% speedup** compared to Gated-DeltaNet in forward propagation, which is of high practical value. Comba has already been open-sourced and supports verification. This performance gain primarily originates from the structural difference in how Comba constructs the **WY representation** (Equation 7), which leads to the **elimination of one matrix inversion** in the UT transformation process (Equation 10). Specifically:
>
> **Gated-DeltaNet**: $M_1 = I-\text{lower}(\text{Diag}(\beta)KK^T)^{-1}, ~~~~M_2 = I-\text{lower}(\text{Diag}(\beta)(\Gamma\odot KK^T))^{-1},~$
> where $\Gamma_{ij}=\prod_{t=1}^j \alpha_t/\prod_{t=1}^i \alpha_t$.
>
> **Comba**: $M = I-\text{lower}(\text{Diag}(\beta)(\Gamma\odot KK^T))^{-1},~$
> where $\Gamma_{ij}=\prod_{t=1}^{j-1} \alpha_t/\prod_{t=1}^i \alpha_t$ **(The numerator contains a shift)**.
>
>
> Matrix inversion is a major performance bottleneck for kernel operators. In models like Comba and Gated-DeltaNet, the matrices to be inverted are masked with lower-triangular patterns, breaking symmetry and preventing the use of the Woodbury identity to reduce complexity to $\mathcal{O}(n^2)$. As a result, inversion typically relies on Gaussian elimination with $\mathcal{O}(n^3)$ complexity. By eliminating one matrix inversion, Comba achieves substantial speed gains.
>
> Importantly, this benefit can be extended to other DeltaNet-style models. Last month, inspired by Comba, the implementation of the open-sourced Gated-DeltaNet was updated: it now first derives a Comba-like formulation and then computes using the Comba operator. **Comba is, to our knowledge, the first model to introduce this operator structure**. This contribution was not emphasized in our original submission but will be further clarified in the revised version.
>
> # W1-W2: The Closed-loop Control in Comba.
>
> Thank you for your valuable feedback. Comba adopts the fundamental definition of closed-loop systems as a **design intuition**. As stated on Wikipedia [8]:
>
> > *A closed-loop controller is a control loop which incorporates feedback. A closed-loop controller uses feedback to control states or outputs of a dynamical system.*
>
> Following this definition, consider the standard RNN update: $S_t = S_{t-1} + v_t k_t^T$. Here, $S_{t-1}$ passively receives the update and does not influence the update behavior through feedback. Therefore, models like **Mamba2** can be viewed as **open-loop systems**.
>
> In contrast, **Comba** introduces two explicit forms of feedback:
>
> - **State correction:** $S_t = S_{t-1} (\alpha_t - k_t k_t^T) + v_t k_t^T$
>
> - **Output correction:** $o_t = S_t q_t + d(v_t - S_t k_t) = S_t(q_t - dk_t)$ (Residual connection $d * v_t$ is omitted, as is also done in Mamba2.)
>
> These mechanisms correspond directly to the two parts of the closed-loop definition: feedback for both **states** and **outputs**. In comparison, models like **RWKV-7** and **DeltaNet** only implement feedback at the **state** level and thus cannot be considered end-to-end closed-loop systems. Empirically, we observe that introducing output correction significantly boosts model performance. Notably, this mechanism can be implemented with a **single line of code** `q_t - d*k_t`, and can be easily integrated into many RNN variants.
>
> Since the design of Comba is primarily intuition-driven, it is difficult to establish a one-to-one correspondence between its components and those of classical control systems. Moreover, if the output correction were to directly influence the state $\mathbf{S}_t$, it would introduce additional nonlinearity that compromises Comba's ability to support hardware-efficient **blockwise parallel computation**—a critical requirement for scaling to large language models.
>
> Due to space limitations, we kindly refer you to our response to reviewer **ScfA** (second to last), specifically in the section titled **"Q2: How does the feedback improve the performance?"**, where we provide a comprehensive explanation of how the feedback mechanisms in Comba contribute to its success from multiple perspectives.
>
> [8] Wikipedia: Closed-loop controller
>
> # W3: Initial value for d.
>
> In our initial experiments, we consistently observed that setting $d = 1$ resulted in the lowest loss. However, during downstream task evaluation with the 340M-scale model, we occasionally found that $d = 0.02$ yielded better results. Out of scientific rigor, we reported this observation faithfully in the original manuscript.
>
> During the review period, we conducted a more detailed investigation into the parameterization of $d$, and we would like to share our latest findings here. The new results confirm that $d = 1$ provides the best performance overall. We believe the earlier deviation was due to the relatively small size of the 340M model, where certain patterns are more prone to noise or instability.
> | Acc. in Commonsense Reasoning|340M/15B (seed=42)| 340M/15B (seed=2025)| 760M/50B | 1.3B/100B |
> |-|-|-|-|-|
> | `w/o. q - dk`|42.45|42.98|45.82|49.72|
> | `d=0.02`|43.36|43.40|45.66|51.78|
> | `d=0.5`|**43.71**|43.37|46.39|51.62|
> | `d=1`|43.26|**43.81**|**46.70**|**51.91**|
>
> Besides, we find that integrating this mechanism into other modern RNNs, such as Gated-DeltaNet and GLA, led to notable loss reductions. (1.3B/100B Gated-DeltaNet dropped from 2.213 to 2.175, and 1.3B/100B GLA dropped from 2.232 to 2.204)
> > We will incorporate all the details into the revised version of the manuscript. If your concerns have been addressed, we would be sincerely grateful if you would consider raising the score. Should you have any further questions, we would be more than happy to respond.

---

> ### Comment · Reviewer_fcuH · 2025-08-05
>
> Thanks to the authors for their detailed response, addressing most of my concerns and questions. I still have one minor concern regarding the usage of the term ‘closed-loop control’, which implies that there is an error calculation mechanism and the error is explicitly minimized through feedback. Although I understand that establishing an one-to-one correspondence between Comba and a classical closed-loop control system is not the goal of the work, I suggest that the authors clarify this explicitly to avoid possible confusion and, ideally, replace ‘closed-loop control’ with another term.

---

> > ### Author Response · Authors · 2025-08-05
> > **Thank you for your response!**
> >
> > Thank you for your further comments. Since Comba's state correction mechanism can also be written as
> >
> > $S_t = \alpha_t S_{t-1} + (v_t - S_{t-1}k_t)k_t^T$
> >
> > which can be understood as minimizing the error between the retrieved past values $S_{t-1}k_t$ and new values $v_t$, it is indeed a closed-loop way of thinking. The output correction technique also explicitly minimizes the error between system queries and keys by $q_t - d k_t$ to facilitate accurate key-value association memory retrieval.
> >
> > We really appreciate your suggestion, and in order to prevent any unnecessary misunderstanding, we will remove the concept of "closed-loop control" from the revised version of the paper and change the title to "Improving Bilinear RNNs with Output Correction." Where the term 'bilinear' refers to the bilinear term introduced by the product of $Sk$ in Comba, which is not present in earlier linear RNNs such as Mamba. Output correction specifically refers to the `q - dk` term. This only requires corresponding changes in Section 3 (lines 144-155) and does not affect other parts of the paper (since the closed-loop control concept is only mentioned in this section).
> >
> > In the revised version, we will clearly state that Comba is designed using the feedback concept from control systems, rather than strictly aligning with classical control systems. Thank you once again for your suggestions, and we wish you all the best.
> >
> > > We do have a small request: if your concerns have been addressed, we kindly ask if you could raise the score to 5. We would be extremely grateful.

---

### Official Review · Reviewer_cuva · 2025-07-02

**Clarity:** 3
**Significance:** 2
**Originality:** 2
**Rating:** 5
**Confidence:** 5

**Summary:**

The paper introduces Comba as a new model to the series of sub-quadratic architectures like Mamba-2 \[1] and Gated-DeltaNet \[2]. It makes the following contributions:

1. It improves the Gated DeltaNet recurrence,

   $$
    \mathbf{S_t}  = \mathbf{S_{t-1}} {\alpha_t} (\mathbf{I} - {\beta_t} \mathbf{k_t}\mathbf{k_t}^T) + \beta_t \mathbf{v_t}\mathbf{k_t}^T
   $$

   to

   $$
     \mathbf{S_t}  = \mathbf{S_{t-1}} (\alpha_t \mathbf{I} - \tilde{\beta}_t \mathbf{k_t}\mathbf{k_t}^T) + \beta_t \mathbf{v_t}\mathbf{k_t}^T
   $$

   untying the two $\beta$s and relocating $\alpha$ (motivated by control theory).

2. It uses UT transforms for its kernel, making it 40 % more efficient than the WY transform.

3. It shows improvements in performance on language modeling, recall tasks, Long Bench, and ImageNet classification.


-----

[1]: Transformers are SSMs: Generalized Models and Efficient Algorithms Through Structured State Space Duality. Tri Dao, Albert Gu

[2]: Gated Delta Networks: Improving Mamba2 with Delta Rule. Songlin Yang, Jan Kautz, Ali Hatamizadeh

**Questions:**

Please see weaknesses

**Ethical Concerns:**

["NO or VERY MINOR ethics concerns only"]

**Final Justification:**

The authors have addressed my concerns on the both the novelty in the use of one inversion v/s two inversions in Gated Deltanet as well as the differences in perplexity reported compared to previous papers. I have raised my score to an accept.

**Limitations:**

Please see weaknesses

**Quality:**

2

**Strengths And Weaknesses:**

### Strengths

1. The paper is well written and the proposed change is straightforward to follow.
2. The PPL improvements on Wiki and LAMBADA over Gated DeltaNet are substantial.
3. The kernel achieves a 40 % speedup compared to the WY transform.

### Weaknesses

1. The switch from WY to UT transform for Delta Rule–based methods appears to have been introduced in the DeltaNet paper \[3]. Could the authors clarify how Comba’s algorithm differs from \[3] beyond the expected updates to the $W$ and $T$ matrices?

2. In Table 6 (main LM results), I think important baselines seem to be under-tuned/missing:

   * **Gated DeltaNet**:

     * DeltaNet \[3] reports Wiki/LAMBADA PPLs of $(28.24\,37.37)$ and $(16.87\,12.21)$ for the 340M/1.3B models on SlimPajama (15 B/100 B tokens).
     * The authors report $(26.47\,45.46)$ and $(17.14\,18.80)$ for Gated DeltaNet. These results are worse than DeltaNet itself, even though Gated DeltaNet is known to more performant than DeltaNet.
     * Comba achieves $(24.15\,39.91)$ and $(16.01\,12.68)$.

     Under this light, the results become much more mixed than they seem to be. Could the authors please comment on this discrepancy

   * **Mamba-2 vs. Mamba**:
     I recommend using Mamba-2 \[1] rather than the original Mamba \[4] in Table 6, as Mamba-2 is the more recent and better-performing variant.

   * **Transformer++**:
     I understand that the authors have used a previous paper for Transformer++ results, but I would recommend training this model in-house. The reported PPL numbers strongly suggest that this baseline has been poorly tuned. For reference, please see that the gap is much less pronounced in the numbers reported in Table 2 in MesaNet \[5]


To sum up, I believe Gated DeltaNet, Mamba-2, and Transformer++ are the most important baselines, and they appear under-tuned/missing. I would recommend better-tuning these baselines to more reliably understand Comba’s improvements.

----

[3]: Parallelizing Linear Transformers with the Delta Rule over Sequence Length. Songlin Yang, Bailin Wang, Yu Zhang, Yikang Shen, Yoon Kim

[4]: Mamba: Linear-Time Sequence Modeling with Selective State Spaces. Albert Gu, Tri Dao

[5]: MesaNet: Sequence Modeling by Locally Optimal Test-Time Training. Johannes von Oswald, Nino Scherrer, Seijin Kobayashi, Luca Versari, Songlin Yang, Maximilian Schlegel, Kaitlin Maile, Yanick Schimpf, Oliver Sieberling, Alexander Meulemans, Rif A. Saurous, Guillaume Lajoie, Charlotte Frenkel, Razvan Pascanu, Blaise Agüera y Arcas, João Sacramento

---

> ### Author Rebuttal · Authors · 2025-07-30
>
> Thank you very much for your detailed review. Your concerns align closely with the clarifications we have incorporated into the revised version. As NeurIPS does not allow document revisions, we will first restate each point and then provide our response.
>
>
> # Q1: The difference between Comba and Gated-DeltaNet.
>
> Comba offers three key contributions compared to Gated-DeltaNet, which we will analyze point by point. Additionally, during the review period, we conducted further experiments, and we would like to share some new findings with you alongside our responses.
>
>
> ## A1.1: Kernel Speedup via Comba's Specific Chunkwise Parallelism
>
> We believe the primary contribution of Comba lies in its direct acceleration of DeltaNet-style operators. Specifically, Comba achieves up to a **46% speedup** compared to Gated-DeltaNet in forward propagation when `head_dim = 128`, which is of high practical value. Comba has already been open-sourced and supports verification.
>
> This performance gain primarily originates from the structural difference in how Comba constructs the **WY representation** (Equation 7), which leads to the **elimination of one matrix inversion** in the UT transformation process (Equation 10). Specifically:
>
> **Gated-DeltaNet**: $M_1 = \mathbf{I}-\text{lower}(\text{Diag}(\beta)\mathbf{K}\mathbf{K}^T)^{-1}, ~~~~M_2 = \mathbf{I}-\text{lower}(\text{Diag}(\beta)(\mathbf{\Gamma}\odot\mathbf{K}\mathbf{K}^T))^{-1},~$
>
> where $\mathbf{\Gamma}_{ij} = \prod_1^j \alpha_t / \prod_1^i \alpha_t$.
>
> **Comba**：$M = \mathbf{I}-\text{lower}(\text{Diag}(\beta)(\mathbf{\Gamma}\odot\mathbf{K}\mathbf{K}^T))^{-1},~$
>
> where $\mathbf{\Gamma}_{ij}=\prod_0^{j-1} \alpha_t/\prod_1^i \alpha_t$  **(The numerator contains a shift)**.
>
>
> Matrix inversion is a major performance bottleneck for kernel operators. In models like Comba and Gated-DeltaNet, the matrices to be inverted are masked with lower-triangular patterns, breaking symmetry and preventing the use of the Woodbury identity to reduce complexity to $\mathcal{O}(n^2)$. As a result, inversion typically relies on Gaussian elimination with $\mathcal{O}(n^3)$ complexity. By eliminating one matrix inversion, Comba achieves substantial speed gains.
>
> Importantly, this benefit can be extended to other DeltaNet-style models. Recently, inspired by Comba, the implementation of the open-sourced Gated-DeltaNet was updated: it now first derives a Comba-like formulation and then computes using the Comba operator. **Comba is, to our knowledge, the first model to introduce this operator structure**. This contribution was not emphasized in our original submission but will be further clarified in the revised version.
>
> ## A1.2: Superior SPLR Structure in Comba
>
> As described in lines 167–181 of the paper, Comba employs an SPLR structure $(\alpha_t-b*\mathbf{k}_t\mathbf{k}_t^{T})$ with eigenvalues in $( -1, 1 )$, which offers strictly greater expressiveness than the $( 0, 1 )$ range used in Gated-DeltaNet $\alpha_t(\mathbf{I}-\mathbf{k}_t\mathbf{k}_t^{T})$. Detailed experiments in Table 3 further demonstrate that among the three latest generation (Delta-based) RNNs that support efficient parallelism (Comba, Gated-DeltaNet, and RWKV-v7), Comba's SPLR structure outperforms the IPLR structure of Gated-DeltaNet and the DPLR structure of RWKV-v7.
>
> Moreover, as shown in Table 3, the update rule used in Gated-DeltaNet appears to be incorrect, as it cannot be explicitly written in the standard Stochastic Gradient Descent (SGD) form. In contrast, Comba's formulation adheres to this structure, making it theoretically sound. This issue has also been noted in several recent blogs (in accordance with NeurIPS rebuttal guidelines, we can not include external links):
>
> **Gated-DeltaNet**: $S_t = S_{t-1} - \beta_t \nabla_S\|\| \frac{1}{ \alpha_t } v_t - \alpha_t S_{t-1} k_t \|\|^2$
>
> **Comba**: $S_t = S_{t-1} - \beta_t \nabla_S\|\| v_t - S_{t-1}k_t \|\|^2$
>
> ## A1.3: Significant Impact of Comba's Output Correction $\mathbf{o}_t=\mathbf{S}_t(\mathbf{q}_t-d\mathbf{k}_t)$
>
> During the review period, we conducted a more comprehensive investigation into Comba’s output correction mechanism (`q_t - d*k_t`) and observed unexpectedly strong improvements. Integrating this mechanism into both the Gated-DeltaNet and GLA models led to notable loss reductions. (1.3B/100B Gated-DeltaNet dropped from 2.213 to 2.175, and 1.3B/100B GLA dropped from 2.232 to 2.204). The significant improvement of Comba over Gated-DeltaNet can also be largely attributed to the output correction mechanism.
>
> While prior RNN research has largely focused on improving gating mechanisms or optimizing state updates via key-value operations, few models have explicitly modified the output pathway (i.e., the query @ State). We think there appears to be a potential connection between Comba and MesaNet, as MesaNet also performs query correction in the output stage by solving a closed-form recursive least squares problem, resulting in $q_t=(H_t + \Lambda)^{-1}q_t$. Notably, the correction in Comba can be implemented **with a single line of code** `q_t - d*k_t` and is applicable to nearly all RNN variants.
>
>
> # Q2: The evaluation result.
>
> Thank you again for your careful and rigorous review. Some experimental details may have caused confusion, and we would like to address each of your concerns regarding evaluation metrics in detail. All experiments are reproducible using the **open-sourced Comba model** and the provided **experiment settings** in the paper.
>
>
> ## A2.1: PPL Gap on Wiki/LAMBADA Compared to DeltaNet
> As shown in Table 6, the Wiki PPL scores in our paper are relatively close to the published DeltaNet paper. We assume your main concern lies with the LAMBADA results.
>
> This discrepancy stems from the fact that the `lm-evaluation-harness` library includes two versions of the LAMBADA dataset: `LAMBADA_openai` and `LAMBADA_standard`. DeltaNet uses `LAMBADA_openai`, while Comba (due to IP restrictions to HuggingFace) and another paper in this field [1] adopt `LAMBADA_standard`. This difference in dataset versions leads to noticeable variation in PPL scores. Thank you for pointing this out. We will clarify which version of the LAMBADA dataset was used in the revised paper.
>
> To address your concern more thoroughly, as shown below, we re-evaluated all models using the `LAMBADA_openai` dataset. The results align with the conclusion reported in our original paper. We think that `LAMBADA_openai` may be a truncated version of `LAMBADA_standard`. Typically, when the input length during evaluation exceeds the pretraining context length, it can lead to a significant increase in PPL. In our experiments with `LAMBADA_standard` using `lm-evaluation-harness`, we found that setting `max_length=2048` during evaluation noticeably reduces the PPL. This further supports our assumption and explains the observed differences.
>
> | PPL in Lambda_openai | 340M/15B |1.3B/100B
> |-|-|-|
> | Transformer++  | 37.78 |  18.37  |
> | Gated-DeltaNet | 34.22  |   17.21   |
> | Comba | **31.17**|  **13.06**  |
>
> [1] Mom: Linear sequence modeling with mixture-of-memories
>
> ## Q2.2 Evaluation for Mamba2.
>
> We appreciate your comment. As prior works [2,3,4] have already demonstrated that Gated-DeltaNet outperforms Mamba2 in fair settings, we chose Gated-DeltaNet as our primary comparison target. We have now added Mamba2 evaluation results for 340M and 1.3B models:
>
> | PPL in Wiki | 340M/15B |1.3B/100B |
> |-|-|-|
> | Mamba2  | 26.94 | 17.31 |
> | Gated-DeltaNet  | 26.47 | 17.14 |
> | Comba | **24.15** | **16.01** |
>
> | Acc. in Commonsense Reasoning | 340M/15B |1.3B/100B |
> |-|-|-|
> | Mamba2  | 42.14 |  49.26 |
> | Gated-DeltaNet  | 42.21 |  49.58  |
> | Comba | **43.36** | **51.91** |
>
> It is clear that Comba > Gated-DeltaNet > Mamba2 across scales. This is likely because Mamba2 still follows the Hebbian learning rule (L1 Loss for state updates), whereas Gated-DeltaNet and Comba rely on the Delta rule (L2 Loss for state updates), which is stronger. We appreciate your suggestion and will include Mamba2 results in the revised paper. And in Table 6, we have already shown that applying a Mamba2-like architecture to Comba leads to a significant performance drop.
>
> [2] Gated delta networks: Improving mamba2 with delta rule
>
> [3] MesaNet: Sequence Modeling by Locally Optimal Test-Time Training
>
> [4] Log-Linear Attention
>
>
> ## Q2.3 Bad PPL for Transformer++.
>
> Thank you for your question. The reason behind the higher PPL for Transformer++ at the 340M scale is again related to the use of `LAMBADA_standard` instead of `LAMBADA_openai`. The results in Answer 2.1 confirm that the models in our paper were not poorly tuned, and the PPL difference arises solely from the evaluation dataset choice. Moreover, all models in the original Comba paper used `LAMBADA_standard`, so the reported comparisons remain fair.
>
> > We will incorporate all the details into the revised version of the manuscript. Considering that **other reviewers have generally regarded this paper as a good submission**, we would be sincerely grateful if you would consider raising the score, provided your concerns have been addressed. Should you have any further questions, we would be more than happy to respond.

---

> > ### Author Response · Authors · 2025-08-07
> > **Sincerely ask whether the concern have been addressed**
> >
> > Hi Reviewer cuva,
> >
> >
> > thank you very much for your careful review. We have now addressed in detail the two points you raised:
> >
> > >(1) the distinctions between Comba and Gated-DeltaNet
> >
> > >(2) the PPL-evaluation differences.
> >
> >
> > Regarding your first point, we have provided a detailed explanation of the distinctions between Comba and Gated-DeltaNet. As for the second point, we have clarified that the PPL discrepancy arises from our dataset choices, not from any limitation of the model itself. To further support this, we have included additional experimental results with Mamba2 that corroborate our conclusion.
> >
> > **With the rebuttal period approaching its end, could you please let us know whether our responses have addressed your concerns? If they have, and in light of the positive evaluations from the other reviewers, we would greatly appreciate your consideration in raising the score.**
> >
> >
> > Should you have any further questions, please continue the discussion and we will respond promptly before the final deadline.
> >
> >
> >
> > Thank you again for your time and help.
> >
> >
> >
> > Best regards,
> >
> >
> > Authors

---

> ### Author Response · Authors · 2025-08-06
> **Sincere Thanks to Reviewer cuva**
>
> Thank you very much for taking the time to review our paper. As the rebuttal period is drawing to a close, I would like to ask whether all your concerns have been satisfactorily addressed. If any questions remain, we are more than happy to continue the discussion and provide further clarification.

---

### Comment · Area_Chair_WRY7 · 2025-08-05
**Rebuttal Phase Closing Soon**

Dear Reviewers,

As the author rebuttal phase will close in less than 48 hours, we kindly ask you to make use of the remaining time to engage with the authors if there are any outstanding questions or concerns. At a minimum, please acknowledge that you have read the rebuttal. If you have already done so, we sincerely appreciate your efforts.

Thank you in advance for your continued contributions.

Best,

AC

---

### Note · Authors · 2025-08-12

We would like to sincerely thank all reviewers for their high-quality comments, which have helped improve *Comba*. Below, we summarize the main concerns raised by each reviewer and provide brief responses, so that the reviewers as well as the AC/PC can better understand the overall rebuttal process.

# Reviewer cuva
1. **Difference between Comba and Gated-DeltaNet**
   - Addressed from three perspectives: **operator derivation**, **expressive power**, and the unique **output correction mechanism**.
2. **Discrepancy in PPL results compared to other papers**
   - Clarified that the difference is due to using the `Lamba_standard` dataset rather than `Lamba_openai`, and is not caused by the model itself.

# Reviewer fcuH
1. **Computational efficiency of Comba**
   - Provided evaluation results across multiple models for efficiency.
2. **Paper details**
   - Offered clarifications and explanations.
3. **Why Comba is faster than Gated-DeltaNet**
   - Explained in detail from the perspective of operator deployment.
4. **Closed-loop control concept**
   - Clarified the concept and, following the reviewer’s suggestion, will explicitly mark it in the revised version to avoid ambiguity.
5. **Initialization choice for parameter *d***
   - Provided additional experimental results.

# Reviewer ScfA
1. **Closed-loop control concept**
   - Addressed in the same way as Reviewer B’s point (4).
2. **Output correction mechanism**
   - Explained from the perspectives of neural memory and optimizer design.

# Reviewer npnS
1. **Comba’s position in the deep learning architecture landscape**
   - Analyzed Comba and other advanced architectures in terms of expressive power and efficiency, starting from the circuit complexity perspective in theoretical computer science.

**Finally, we would like to once again thank all reviewers and the AC/PC for their time and effort in reviewing this paper.**

---

### Decision · Program_Chairs · 2025-09-17

**Decision:**

Accept (spotlight)

**Comment:**

This paper introduces Comba, a new nonlinear RNN architecture inspired by control theory. Reviewers agreed that the paper is very well written, and well motivated by prior work. This paper leverages the recent developments of nonlinear RNNs and demonstrates some of their advantages. The empirical results are extensive, covering language modeling, reasoning, long-context benchmarks, and vision tasks, with Comba showing consistent improvements. The ablation studies are detailed. And the release of code enables reproducibility and transparency.  Some minor weaknesses were noted: the control-theoretic framing could be clarified (e.g., with block diagrams); some terminology may be confusing; some of the baselines should be better tuned for fairness. All of these issues can be addressed in the camera ready version.

Overall, this is a strong paper that provides both methodological novelty and convincing empirical evidence. Investigating nonlinear RNNs provides a fresh perspective and new impulses, which I think is important. This paper is a strong accept.